

# Identifying uncertainties in simulated streamflow from hydrologic model components for climate change impact assessments

Dongmei Feng[1*] and Edward Beighley[2,3]

[1] Civil and Environmental Engineering, University of Massachusetts, Amherst, MA, USA

[2] Civil and Environmental Engineering, Northeastern University, MA, USA

[3] Marine and Environmental Sciences, Northeastern University, MA, USA

[*]Corresponding author, email address: dmei.feng@gmail.com, telephone: (617) 697-8789

**Abstract:** Assessing the impacts of climate change on hydrologic systems is critical for developing adaptation and mitigation strategies for water resource management, risk control and ecosystem conservation practices. Such assessments are commonly accomplished using outputs from a hydrologic model forced with future precipitation and temperature projections. The algorithms used in the hydrologic model components (e.g., runoff generation) can introduce significant uncertainties in the simulated hydrologic variables, yet the identification and quantification of such uncertainties is rarely studied. Here, a modeling framework is developed that integrates multiple runoff generation algorithms with a routing model and associated parameter optimizations. This framework is able to identify uncertainties from both hydrologic model components and climate forcings as well as associated parameterization. Three fundamentally different runoff generation approaches: runoff coefficient method (RCM, conceptual), variable infiltration capacity (VIC, physically-based, infiltration excess) and simple-TOPMODEL (STP, physically-based, saturation excess), are coupled with Hillslope River Routing model to simulate streamflow. A case study conducted in Santa Barbara County, California, reveals that the median changes are 1-10% increases in mean annual discharge ($Q_m$) and 10-40% increases in annual maximum daily discharge ($Q_p$) and 100-yr flood discharge ($Q_{100}$). The Bayesian Model Averaging analysis indicates that the probability of increase in streamflow can be up to 85%. However, the simulated discharge uncertainties are large (i.e., 230% for $Q_m$ and 330% for $Q_p$ and $Q_{100}$) with general circulation models (GCMs) and emission scenarios accounting for more than half of the total uncertainty. Hydrologic process models contribute 10-30% of the total uncertainty, while uncertainty due to hydrologic model parameterization is almost negligible (<1%), limiting the impacts of hydrologic model parameter equifinality in climate change impact analysis. This study also provides insights on how to optimize the selection of hydrologic models for projecting future streamflow conditions.



## 1. Introduction

Streamflow is essential to humans and ecosystems, supporting human's life and economic activities, providing habitat for aquatic creatures, and exporting sediment/nutrients to coastal ecosystems (Feng et al., 2016;Barnett et al., 2005;Milly et al., 2005). Understanding streamflow characteristics is important for water-resources management, civil infrastructure design and making adaptation strategies for economic and ecological practices (Feng et al., 2019). With economic development and population growth, the emission of greenhouse gas is likely to increase during 21$^{st}$ century (IPCC, 2014). The increase in global surface temperature is projected to exceed 2°C by the end of 21$^{st}$ century even under moderate emission scenarios (e.g., Representative Concentration Pathways, RCPs, 4.5 and 6.0) (IPCC, 2014). Intensified hydro-meteorological processes, altered precipitation forms and patterns, and intensified atmospheric river events and oceanic anomalies (e.g. El Nino events) are projected and likely to causes substantial impacts on hydrologic fluxes (e.g., streamflow) (Barnett et al., 2005;Tao et al., 2011;Dai, 2013;Dettinger, 2011;Vicky et al., 2018;Cai et al., 2014;Feng et al., 2019).

The integration of climate projections and hydrologic models enables the investigation of streamflow dynamics under the future climate conditions. However, the simulated streamflow contains uncertainties from various sources. Due to epistemic limitations (e.g., human's lack of knowledge about hydrologic processes and boundary conditions) and the complexities in nature (e.g., temporal and spatial heterogeneity), hydrologic models are simplified representations of natural hydrologic processes (Beven and Cloke, 2012). Generally, hydrologic models have modules simulating atmosphere-land interactions associated with water and energy partitioning (named as runoff generation process in this study), and modules simulating the water transportation along terrestrial hillslopes and channels (named as routing process here). Each process can be represented in different ways, which thus results in uncertainties in simulated streamflow. For the runoff generation process, surface runoff is mainly represented as infiltration excess overland flow (or Hortonian flow (Horton, 1933)) or saturation excess overland flow. Infiltration excess overland flow occurs when water falls on the soil surface at a rate higher than that the soil can absorb. Saturation excess overland flow occurs when precipitation falls on completely saturated soils. In addition, surface runoff can also be quantified conceptually, for example, a runoff coefficient can be used to generate surface runoff as a proportion of precipitation



rate. Subsurface runoff is generally represented as functions of soil characteristics and topographic
features. The complexity of these functions varies significantly, from simple linear to
combinations of multiple non-linear. The lateral routing processes are generally represented using
various approximations of the Saint-Venant equations (Reed et al., 2004). Difference choices of
these process models may achieve different results and thus cause uncertainties in outputs.
Parameterization can be another uncertainty source. Due to the nonlinearity of hydrologic
processes, different combinations of model parameters can achieve similar, if not identical, model
performance. Model parameter selections based on statistical metrics obtained from calibration
can result in different optimal parameter values (i.e., parameter equifinality). When it comes to
hydrologic impact assessments, the model forcings, which differ among General Circulation
Models (GCMs) due to the model discrepancy and the uncertainty of future emission scenarios,
also contribute to the uncertainties in simulated streamflow. Without appropriate assessment of
these uncertainties, standalone studies on the climate change impacts, using a particular hydrologic
model forced by select GCMs' projections under some emission scenarios, can be difficult to
interpret. Systematic assessments of the relevant uncertainties associated with simulated
streamflow are needed.

Some studies have been performed considering the above at both regional and global scales

(for example, (Wilby and Harris, 2006;Vetter et al., 2015;Valentina et al., 2017;Kay et al.,
2009;Eisner et al., 2017;Su et al., 2017;Schewe et al., 2014;Hagemann et al., 2013;Asadieh and
Krakauer, 2017)). Most previous studies integrated multiple hydrologic models individually.
However, the hydrologic model structures can be significantly different, which may limit the
ability to quantify relative uncertainty contributions from different model components (e.g., runoff
generation model and routing model) and associated parametrizations. Troin et al. (2018) tested
the impacts by using different hydrologic model components to simulate streamflow, but they only
focused on snow and potential ET methods. In this study, a consistent hydrologic modeling
framework that integrates multiple runoff generation process models with surface, subsurface and
channel routing processes and associated parameter uncertainties is developed. This framework
enables uncertainties from different components representing hydrologic processes and associated
model parameters as well as model forcings (e.g., precipitation and temperature) to be quantified
and compared in a consistent manner. In this framework, three runoff generation process models
which represent the three fundamentally different approaches mentioned above are used. The





conceptual frameworks are adapted from the variable infiltration capacity model (Wood et al.,
1992;Liang et al., 1996) (infiltration excess), simple-TOPMODEL (Niu et al., 2005) (saturation
excess), and the runoff coefficient method (Feng et al., 2019) (conceptual). Each approach is
coupled within one routing model (i.e., Hillslope River Routing model, HRR (Beighley et al.,
2009)) to investigate the impacts of model structures and associate parameters on simulated
streamflow. Compared to runoff generation process models, routing process models have less
variants with most models using approximations of the Saint-Venant equations (Reed et al., 2004).
Therefore, only one routing model is included in this study, however, this modeling framework is
suitable to integrate different routing process models (e.g., diffusive wave and full dynamic
solutions) and runoff schemes in future studies. This modeling framework is also coupled with a
Bayesian model averaging (BMA) analysis to assess the performance of different model-forcing-
parameter combinations and to provide actionable information (e.g., probability of estimated
changes) for associated practices, such as water resource management and ecology conservation.
A case study is presented for Santa Barbara County, CA, a biodiverse region under a
Mediterranean climate with a mix of highly developed and natural watersheds. To estimate future
streamflow and associate uncertainties, the hydrologic models are forced with climate projections
from 10 GCMs selected for their good performance in representing historical meteorological
characteristics in the study region, under 2 emission scenarios (RCP 4.5 and RCP 8.5) (Feng et al.,
2019).  The main objectives of this study are to: (1) evaluate and compare the performance of
hydrologic models with different approaches representing runoff generation process  using a
consistent modeling framework; (2) quantify the relative contributions of different sources
(including hydrologic process models, parameterizations, GCM forcings and emission scenarios)
to the total uncertainty in simulated streamflow; and (3) provide actionable information and
suggestions for studies and practices associated with the hydrologic impacts of climate change in
the study region.
**2.  Methods**
2.1 Study region
The study region is located in coastal Santa Barbara County (SBC), California, where
watersheds drain into the Santa Barbara Channel from just west of the Ventura River to just east





of Point Conception (Fig. 1). The combined land area is roughly 750 km2 with 135 watersheds
ranging from 0.1 to 123 km2. The local climate is Mediterranean, with an average annual
precipitation of roughly 600 mm (Feng et al., 2019). Most of the annual precipitation occurs in
fall/winter with 85% of rainfall occurring in the November-March period. Thus, it is characterized
by the intense and flashy floods in winter time. More than 80% of annual discharge occurs in only
a few number of large events during January-March and a large fraction of annual discharge
happens within one day (Beighley et al., 2003). River channels are typically filled with sediment
during dry season (April-October) and are scoured with the initiation of wet season floods (Scott
and Williams, 1978;Keller and Capelli, 1992). River flow is the major source of sediment exported
to the coastal sandy beaches in SBC. Therefore, the timing of seasonality and magnitudes of flood
events are critical to both local community and coastal ecosystems.
2.1 Data

Daily precipitation and temperature with a spatial resolution of 0.0625° x 0.0625° (roughly

6 by 6 km) (Livneh et al., 2015), and daily streamflow from 4 USGS gauges for the period 1984-
2013 are used to calibrate and validate the hydrologic model. The Global Soil Dataset for use in
Earth system models (GSDE) is used to estimate saturated hydraulic conductivity and saturated
moisture content. The 16-day composite albedo product (MCD43C3) with a spatial resolution of
0.05° x 0.05° and the monthly aerosol optical depth product (MOD08M3) with a spatial resolution
of 1.0° x 1.0° both derived from NASA's Moderate Resolution Imaging Spectroradiometer
(MODIS) are used to determine net radiation for evapotranspiration (PET) estimation.

For the historical (1986-2005) and future climate simulations (2081-2100), downscaled

precipitation and temperature from ten climate models (please refer to Pierce et al. (2014) and
Pierce et al. (2015) for model details) in Coupled Model Inter-Comparison Project, Phase 5,
(CMIP5) (Taylor et al. 2012) for two emission scenarios RCP 4.5 and RCP 8.5 (Moss et al. 2010)
are used.  These 10 GCMs are selected because they have the best performance in representing
historical climate dynamics at southwest U.S. and California state scales (Pierce et al., 2018).
2.2 Hydrologic modeling framework
2.2.1   Hydrologic model development



This modeling framework is developed on the basis of the Hillslope River Routing model
(HRR) (Beighley et al., 2009). The watershed is delineated based on the Digital Elevation Model
(DEM). The sub-basins are irregular-shape catchments defined by the flow accumulation area
threshold. In this study, the threshold is 1 $km^2$, which means the sub-basins (model units) are in
size of roughly 1 $km^2$. The hydrogeological model inputs, including surface roughness, saturated
hydraulic conductivity, soil thickness, porosity, plane slope, channel slope and channel roughness,
are averaged over each sub-basin. The geometry of each sub-basin (plane length and width) is
calculated based on an "open-book" assumption, which assumes each sub-basin is a rectangular
divided by the river channel into two identical parts like an open book. Please refer to Beighley et
al. (2009) for more details. The precipitation and ET are extracted from the grid-based datasets for
each sub-basin using an area-weighted average method. Then the water-balance model (i.e., runoff
generation method) is applied to each model unit to simulate runoff generation processes. Here,
three runoff generation methods: runoff coefficient (Feng et al., 2019), and the methods used in
Variable Infiltration Capacity (VIC) (Wood et al., 1992;Liang et al., 1996) and simple-
TOPMODEL model (Niu et al., 2005), are used to simulate the generation of surface and
subsurface runoff excess. The routing methods within the HRR model (i.e., kinematic wave for
surface and subsurface lateral routing and Muskingum-Cunge for channel routing) are used to
simulate the transport of runoff excess. To clarify, we denote the three runoff generation
algorithms: runoff coefficient, runoff generation method used in Variable Infiltration Capacity and
runoff generation method used in simple-TOPMODEL as RCM, VIC and STP, respectively. The
three hydrologic models which integrate each of these runoff generation methods with the routing
method used in HRR model are referenced as RCM-HRR, VIC-HRR and STP-HRR, respectively.
The differences between simulations from these three models are considered as the uncertainty
resulting from hydrologic model formulation. The three runoff generation algorithms, and the
surface, subsurface and channel routing are described below.
The RCM assumes water excess available for surface runoff ($e_s$) is proportional to
precipitation rate (P). The proportion is represented by a coefficient value (e.g., 0 to 100%) and is
dependent on land cover, soil and topographic characteristics. The coefficient value is smaller for
dry and flat areas with permeable soils and vegetated surfaces, as compared to that for wet and
steep areas with more impervious areas (e.g., roads, parking lots, roofs).  In this work, a dual
runoff-coefficient method is used, which assigns a larger runoff coefficient ($C_2$) to wet soils





(relative soil moisture at upper soil layer $\theta_U \geq$ threshold $\theta_t$) and smaller runoff coefficient ($C_1$) to
dry soils (relative soil moisture $\theta_U <$ threshold $\theta_t$) (Eq. (1)). The water excess available for
subsurface runoff ($e_{ss}$) is a function of saturated hydraulic conductivity ($k_{sat}$) and relative soil
moisture in lower soil layer ($\theta_L$) (Eq. (2)).

$$
\begin{aligned}
e_s &= C_1 \times P \quad for\ \theta_U < \theta_t \\
&= C_2 \times P \quad for\ \theta_U \geq \theta_t
\end{aligned}
\tag{1}
$$

$$
e_{ss} = K_{sat\_all} k_{sat} \times \left(\frac{\theta_L}{n}\right)^b
\tag{2}
$$

where $e_s$ and $e_{ss}$ are water excess available for surface and subsurface runoff, respectively, (m d$^{-1}$
$^{1}$); P is precipitation rate (m d$^{-1}$); $C_1$ is dry runoff coefficient; $C_2$ is wet runoff coefficient; $\theta_U$ and
$\theta_L$ are relative soil moisture at upper and lower soil layer, respectively; $\theta_t$ is relative soil moisture
threshold differentiating dry and wet soil conditions; $k_{sat}$ is saturated hydraulic conductivity (m
d$^{-1}$); $K_{sat\_all}$ is a scaler; b is Clapp-Hornberger parameter and n is soil porosity. $C_1$, $C_2$, $\theta_t$ and
$K_{sat\_all}$ are parameters needing calibration.

In the VIC algorithm, surface runoff is generated as infiltration excess where the infiltration

rate is characterized by the variable infiltration curve (Wood et al., 1992). In this work, the
framework of modified 2-layer VIC model (VIC-2L) (Liang et al., 1996) is used. The water excess
available for surface runoff is calculated as shown in Eq. (3)-(4). The water excess available for
subsurface runoff is a function of soil moisture in lower soil layer (Eq. (5)), which is a linear
function of soil moisture when the soil is relatively dry and quadratic when the soil is close to
saturation:

$$
e_s = P - z(\theta_s - \theta_U)/\Delta t - z\theta_s \left(max\left|0, \left[1 - \frac{i_o + P\Delta t}{i_m}\right]\right|\right)^{1+b_i} / \Delta t
\tag{3}
$$

$$
i_o = i_m\left[1 - (1 - A)^{1/b_i}\right]
\tag{4}
$$

$$
e_{ss} = \frac{D_S D_M}{W_S \theta_S}\theta_L + \left(D_M - \frac{D_S D_M}{W_S}\right)\left(\frac{max|0, \theta_L - W_S\theta_S|}{\theta_S - W_S\theta_S}\right)^2
\tag{5}
$$

where z is soil depth in upper layers (m); $\theta_s$ is relative soil moisture at saturation; $i_m$ is maximum
infiltration capacity (m); $i_0$ is infiltration capacity (m); $b_i$ is infiltration curve parameter; A is the
fraction of saturation; $D_M$ is maximum base flow (m d$^{-1}$); $D_S$ is the fraction of $D_M$ at which the non-





linear base flow begins; $W_S$ is the fraction of saturation at which the non-linear base flow
occurs; $\Delta t$ is time step (d). $b_i$, $D_M$, $D_S$ and $W_S$ are parameters which need calibration.
In STP algorithm, the surface runoff is generated as saturation excess overland flow (Eq.
(6)). The saturation fraction of the catchment $f_{sat}$ is determined as a function of topographic index
(Eq. (7)-(8)).

$$e_s = f_{sat} * P \qquad (6)$$
$$f_{sat} = f_{max} * \exp(-0.5\, z_\nabla\, f_{over}) \qquad (7)$$

where $f_{sat}$ is the fraction of saturated area; $f_{over}$ is a decay factor for surface runoff water excess
(m$^{-1}$); $z_\nabla$ is groundwater table depth (m); $f_{max}$ is the maximum saturated fraction and is defined
as the percent of grid cells in each sub-basin with a topographic index ($\tau$) that is $\geq$ the mean $\tau$
determined by averaging all grid cell $\tau$ values:

$$\tau = \ln\left(\frac{a}{tan(\beta)}\right) \qquad (8)$$

where $a$ is the specific catchment area (i.e.,upslope area per unit contour length) and $\beta$ is the pixel
slope. The specific catchment area $a$ and slope $\beta$ are calculated for grid cell using the gridded
elevation data and the TauDEM tools (Tarboton, 2003).
The water excess available for subsurface runoff is a function of maximum base flow rate
and groundwater table depth:

$$e_{ss} = Q_m * \exp(-f_{drain} * z_\nabla) \qquad (9)$$

where $f_{drain}$ is a decay factor for subsurface runoff water excess (m$^{-1}$), and $Q_m$ is the maximum
baseflow rate (m d$^{-1}$). Water excess for both surface and subsurface runoff are dependent of the
groundwater table depth $z_\nabla$. Here, the water table depth $z_\nabla$ is determined by applying the method
used in (Niu et al., 2005), which assumes the water head at depth z is in equilibrium with that at
ground water depth $z_\nabla$ (Eq. (10)-(13)).

$$\varphi(z) - z = \varphi_{sat} - z_\nabla \qquad (10)$$



where $\varphi(z)$ and $\varphi_{sat}$ are the metric potentials at depth z and at groundwater table depth $z_\nabla$ (m).
The soil at the groundwater table depth is assumed to be saturated. Based on Clapp-Hornberger
relationship (Clapp and Hornberger, 1978), $\varphi(z)$ can be expresses as:

$$\varphi(z) = \varphi_{sat}(\frac{\theta(z)}{\theta_{sat}})^{-b} \qquad (11)$$

where $\theta(z)$ and $\theta_{sat}$ are soil moisture content at depth z and groundwater table depth $z_\nabla$,
respectively, b is a Clapp-Hornberger parameter. By substituting Eq. (10) with Eq. (11), the soil
matric profile at depth z can be expressed as:

$$\theta(z) = \theta_{sat}(\frac{\varphi_{sat} - (z_\nabla - z)}{\varphi_{sat}})^{-1/b} \qquad (12)$$

Then, the groundwater table depth ($z_\nabla$) can be determined by solving Eq.13 iteratively.

$$D_\theta = \int_0^{z_\nabla} (\theta_{sat} - \theta(z))dz \qquad (13)$$

where $D_\theta$ is the soil moisture deficit, which can be calculated in Eq.14:

$$D_\theta = \sum_{i=1}^{m} (\theta_{sat} - \theta_i)\nabla z_i \qquad (14)$$

where $\theta_i$ is the soil moisture content at the $i^{th}$ soil layer; $\nabla z_i$ is the soil thickness of $i^{th}$ soil layer, m
is the number of soil layer, m=2 in this study. In STP algorithm, $f_{over}$, $f_{drain}$, $Q_m$ and $\varphi_{sat}$ are
parameters to be calibrated.

The water movement between soil layers in the soil matrix is similar to that in the modified

VIC-2L model (Liang et al., 1996). The soil is divided into 2 layers: upper layer (0.6 m) and lower
layer (2.6 m). The soil thickness data is determined based on a previous study (Feng et al., 2019).
After the surface runoff is determined using the methods mentioned previously, the infiltrated
water is added to the upper soil layer, and the soil moisture is updated. If the upper soil is
oversaturated, the excess water is returned to surface. The interaction between upper and lower
soil layers is determined using the Clapper-Hornberger equation (Eq. (15)-(16)). Subsurface runoff
is generated from the bottom of the lower soil layer. The water flux from the upper layer does not





contribute to runoff and is only lost to evapotranspiration and/or drainage to the lower soil layer.
A conceptual illustration of the runoff generation process for each method and the water movement
in soil matrix can be found in *Supporting Information Fig. S1*.

$$K = k_{sat} \times (\frac{\theta_U}{n})^c \tag{15}$$

$$D = k_{sat} \times (\frac{\theta_L}{n})^c \tag{16}$$

where K is the water flux from the upper soil layer to the lower soil layer (m d$^{-1}$); and D is the
water flux transported from the lower soil layer to the upper soil layer due to diffusion (m d$^{-1}$).

After water excess for surface and subsurface runoff is determined, the kinematic wave

approach is used to simulate the transport of runoff from the planes (surface and subsurface), and
the Muskingum Cunge method is used for channel routing following the below conservation
equations (Beighley et al., 2009):
Plane Routing:

$$\frac{\partial y_s}{dt} + \frac{\partial q_s}{dx_p} = e_s \tag{17}$$

$$\frac{\partial y_{ss}}{dt} + \frac{\partial q_{ss}}{dx_p} = e_{ss} \tag{18}$$

Channel Routing:

$$\frac{\partial A_c}{dt} + \frac{\partial Q_c}{dx_c} = q_s + q_{ss} \tag{19}$$

where y$_s$ and y$_{ss}$ are water depth (or thickness) of surface and subsurface runoff, respectively (m);
q$_s$ and q$_{ss}$ are surface and subsurface runoff flow rates per unit width of plane (m$^2$ s$^{-1}$); dx$_P$ is the
distance step along the plane (m); A$_C$ is the cross section area of flow in the channel (m$^2$); Q$_c$ is
the flow rate in channel (m$^3$ s$^{-1}$); dx$_c$ is the distance step along the channel (m); and *dt* is the time
step (s).
2.3.2   Model calibration





After the models are setup, a state-of-the-art optimization algorithm, Borg Multiobjective
Evolutionary Algorithm (Borg MOEA) (Hadka and Reed, 2013), is adopted to optimize the model
parameters (Table 1). The parameters of the three models are calibrated separately. For each
model, there are 4 parameters calibrated for runoff generation processes and 2 parameters
calibrated for routing processes. $K_{s\_all}$ and $K_{ss\_all}$ are conceptual parameters which account for
spatial heterogeneity at the model unit scale and uncertainties in the hydro-geologic inputs
associated with the plane routing processes (e.g., surface roughness and saturated hydraulic
conductivity). They can be different for different model structures even for the same study region.
Therefore, they are calibrated for each model separately. The Nash–Sutcliffe model efficiency
coefficient (NSE) (Eq. (20)) is used to assess model performance, as it accounts for model
performance in terms of both timing and magnitudes of peak flow and base flow that are
particularly important in this study. The optimal parameter set is determined after the improvement
of error is minimized (here it is defined as ΔNSE<0.005). To quantify the uncertainties from model
parameters, 3 optimal parameter sets with similar performance are selected for each model. The
selected parameter sets are then used for simulation with different climate forcings.

$$\text{NSE} = 1 - \frac{\sum_{t=1}^{T}(Q_s^t - Q_o^t)^2}{\sum_{t=1}^{T}(Q_o^t - \overline{Q_o})^2} \tag{20}$$

where $Q_s^t$ and $Q_o^t$ are simulated and observed discharge at time t, respectively, ($m^3\,s^{-1}$); and $\overline{Q_o}$ is
the mean discharge during the study period of length T, ($m^3\,s^{-1}$).
2.3 Uncertainty Analysis
The uncertainty is quantified by running each of the 9 hydrologic model-parameter sets
(i.e., 3 hydrologic models and 3 parameter sets, 3x3 = 9) with each of the 20 forcing sets (i.e., 10
GCMs and 2 emission scenarios, 10x2=20) for a total of 180 simulations.
To evaluate the uncertainty sources and their relative significance in simulated discharges
for the future period, the analysis of variance (ANOVA) (Vetter et al., 2015) is used. The
contribution of each uncertainty source for a particular variable (e.g., annual mean discharge,
annual peak discharge or 100-yr flood discharge) is defined as the fraction of its variance to the
total variance. The total variance is quantified as the total sum of squares ($SS_{total}$) of differences
between the simulations and the mean of all simulations (Eq. (21)):

$$SS_{Total} = \sum_{i=1}^{N_{Hyd}} \sum_{j=1}^{N_{para}} \sum_{k=1}^{N_{GCM}} \sum_{l=1}^{N_{RCP}} (q_{ijkl} - q_{oooo})^2 \qquad (21)$$

where $q_{ijkl}$ is the simulated value of a particular variable by i$^{th}$ hydrologic model with j$^{th}$ parameter
set, forced by k$^{th}$ GCM projection under l$^{th}$ RCP scenario; $q_{oooo}$ is the overall average of the
simulated variable. Next, the SS$_{Total}$ can be divided into 15 parts representing the 4 main effects
(or first-order effects), 6 second-order, 4 third-order and 1 fourth-order interaction effects. For
clarity, the third and fourth orders of interaction effects are combined and represented as SS$_{3.4}$ in
Eq. (22).

$$
\begin{aligned}
SS_{Total} = {} & SS_{Hyd} + SS_{para} + SS_{GCM} + SS_{RCP} + SS_{Hyd.para} + SS_{Hyd.GCM} \\
& + SS_{Hyd.RCP} + SS_{para.GCM} + SS_{para.RCP} + SS_{GCM.RCP} \\
& + SS_{3.4}
\end{aligned}
\qquad (22)
$$

where $SS_{Hyd}, SS_{para}, SS_{GCM}$ and $SS_{RCP}$ are the main effects (i.e., uncertainties or variance) from
hydrologic models, parameterization, GCMs and RCPs, respectively; $SS_{Hyd.para}$, $SS_{Hyd.GCM}$,
$SS_{Hyd.RCP}$, $SS_{para.GCM}$, $SS_{para.RCP}$ and $SS_{GCM.RCP}$ are uncertainties from interactions between
the hydrologic models and parameterization, hydrologic models and GCMs, hydrologic models
and RCPs, parameterization and GCMs, parametrization and RCPs, and GCMs and RCPs,
respectively.  The calculation of the effect of each order is illustrated in Eq. (23)-(25).

$$SS_{Hyd} = N_{para}N_{GCM}N_{RCP} \sum_{i=1}^{N_{Hyd}} (q_{iooo} - q_{oooo})^2 \qquad (23)$$

$$SS_{Hyd.para} = N_{GCM}N_{RCP} \sum_{j=1}^{N_{para}} \sum_{i=1}^{N_{Hyd}} (q_{ijoo} - q_{iooo} - q_{ojoo} + q_{oooo})^2 \qquad (24)$$

$$
\begin{aligned}
SS_{3.4} = {} & SS_{Total} - (SS_{Hyd} + SS_{para} + SS_{GCM} + SS_{RCP} + SS_{Hyd.para} \\
& + SS_{Hyd.GCM} + SS_{Hyd.RCP} + SS_{para.GCM} + SS_{para.RCP} \\
& + SS_{GCM.RCP})
\end{aligned}
\qquad (25)
$$



where $q_{iooo}$ is the average of all simulations from the i$^{th}$ hydrologic model with all combinations
of parameter sets, GCMs and RCPs; $q_{ojoo}$ is the average of all simulations from the j$^{th}$ parameter
set with all combinations of hydrologic models, GCMs and RCPs; $q_{ijoo}$ is the average of all
simulations from the i$^{th}$ hydrologic model and j$^{th}$ parameter set with all combinations of GCMs and
RCPs. Other terms in Eq. (22) can be calculated similarly using Eq. (23)-(24).
To avoid bias from the difference in sample sizes of uncertainty sources (i.e., 3 hydrologic
models, 3 parameter sets, 10 GCMs and 2 RCPs), a subsampling step is performed by following
Vetter et al. (2015). In the subsampling step, 2 samples (the minimum number of uncertainty
source, here it is RCPs) from each source are randomly selected, that is, 2 hydrologic models, 2
parameter sets, 2 GCMs and 2 RCPs, which means $N_{Hyd}$, $N_{para}$, $N_{GCM}$ and $N_{RCP}$ in Eq. (21), (23)-
(24) are all equal to 2. This produces $C_3^2 \times C_3^2 \times C_{10}^2 \times C_2^2$=405 subsamples. For each subsample,
the fractional sum of squares is calculated for each effect using Eq. (23)-(25), and then the average
of variance fractions of each source is used as the uncertainty contribution from that source using
Eq. (26):

$$\delta_e = \frac{1}{405} \sum_{m=1}^{405} \frac{SS_e(m)}{SS_{Total}(m)} \qquad (26)$$

where $\delta_e$ is the average fractional effect of term e (i.e, each of 11 terms in Eq. (22)); $SS_e(m)$ is
the sum of variance of effect e in the m$^{th}$ subsample, and the $SS_{Total}(m)$ is the total variance in
the m$^{th}$ subsample. So in this study, there are 11 $\delta_e$ values in total, representing the uncertainty
contributions of 11 terms in Eq. (22), with a sum of 1.0.
2.5 Probability of estimated changes
In addition to the quantification of uncertainties and associated contributions from different
sources, an evaluation on the probability of uncertain changes in discharge can be useful to provide
actionable information for the stakeholders such as water resource mangagers. In this study, the
Bayesian model averaging (BMA) (Duan et al., 2007) is used to evaluate the model performance
in reproducing historical hydrologic conditions and then assign weights to each of them based on
their performance. A model with better performance will be assigned a higher weight, which
assumes it has a higher probability to be the truth. Note, there is no RCPs for historical period, so
only combinations of hydrologic models, parameter sets and GCMs (3x3x10=90) are evaluated.


Here the models' performance in representing annual mean discharge ($Q_m$) and annual maximum
daily discharge ($Q_p$) is evaluated. The details of this procedure can be found in Chapter 6 in Feng

(2018).

After the weights of model ensemble are obtained using the BMA method, the statistics of
posterior probability distribution (here it is assumed to be normal distribution) of estimated
changes in Qm, Qp and $Q_{100}$ in the future (2081-2100) relative to historical period 1986-2005 are
calculated using Eq. (27)-(32).
For $Q_m$, the statistics are:

$$\mu_m = \sum_{k=1}^{K} w_{k,m} \times c_{k,m} \tag{27}$$

$$\sigma_m{}^2 = \sum_{k=1}^{K} w_{k,m} \times (c_{k,m} - \mu_m)^2 \tag{28}$$

where $\mu_m$ and $\sigma_m$ are the mean and standard deviation of posterior distribution of relative changes
in Qm; $w_{k,m}$ is the weight of model k in terms of Qm; $c_{k,m}$ is the relative change in Qm predicted
by model k; K is the total number of models, and here it is 90.
For Qp, the statistics are:

$$\mu_p = \sum_{k=1}^{K} w_{k,p} \times c_{k,p} \tag{29}$$

$$\sigma_p{}^2 = \sum_{k=1}^{K} w_{k,p} \times (c_{k,p} - \mu_p)^2 \tag{30}$$

where $\mu_p$ and $\sigma_p$ are the mean and standard deviation of posterior distribution of relative changes
in Qp; $w_{k,p}$ is the weight of model k in terms of Qp; $c_{k,p}$ is the relative change in Qp predicted by
model k.
For $Q_{100}$, the statistics are:

$$\mu_{100} = \sum_{k=1}^{K} w_{k,p} \times c_{k,100} \tag{31}$$

$$\sigma_{100}{}^2 = \sum_{k=1}^{K} w_{k,p} \times (c_{k,100} - \mu_{100})^2 \tag{32}$$

where $\mu_{100}$ and $\sigma_{100}$ are the mean and standard deviation of posterior distribution of relative
changes in Q$_{100}$; $w_{k,p}$ is the weight of model k for Q$_P$; $c_{k,100}$ is the relative change in Q$_{100}$
predicted by model k. Here, the weights for Q$_P$ are used because Q$_{100}$ is estimated based on the
statistics of Q$_P$ series, so it is reasonable to assume that the model having a better ability in
reproducing the annual peak discharge should also have a better ability in reproducing the Q$_{100}$.
**3    Results and Discussion**
3.1 Hydrologic model performance
The three hydrologic models perform well in representing streamflow dynamics in the study
region. The NSE varies within 0.57-0.61 and 0.53-0.62 for calibration and validation periods,
respectively, in Mission Creek (gauge NO. 11119750) (Fig. 2). At other calibrated watersheds, the
models perform similarly well with NSE varying between 0.45-0.60 for calibration period and
0.42-0.62 for validation period (Fig. S2-S4). Simulated streamflow from the three models matches
the in-situ measurements in both magnitudes and timing of hydrographs at event scales (Fig. 2b).
At annual scale, simulated annual peak flows are comparable to the observations in most years.
However, in some years with extremely high events, for example in 1995 January, 1998 February
and 2005 January (highlighted in Fig. 2c), the simulated peaks are much lower than the gauge
records. This disparity can be attributed to the input bias (e.g., precipitation or streamflow
measurements). This is identified using an 'extreme scenario' simulation, which assumes 100%
precipitation is transformed to surface runoff (i.e., without any loss due to, for example, infiltration
or evapotranspiration) and transported immediately to river channels and represents the maximum
streamflow considering groundwater is minimal in the study region(Beighley et al., 2003). Even
for this extreme scenario, the simulated peaks were still lower (events highlighted in red in Fig.
2c) or slightly higher (event highlighted in blue in Fig. 2c) than the gauge observations. This is
likely because that model forcings are bias low for these events. One possible source of this bias
can be the grid-based precipitation dataset which averages the precipitation rates over the grid



masking spatial heterogeneity and thus reducing precipitation rates at some locations. The
uncertainties in gauge measurements can also be a bias source. For example, in typical conditions
the uncertainty in streamflow measurements ranges between 6%-19% in small watersheds, but it
can be higher during large storm events when accurate stage measurements are more difficult
(Harmel et al., 2006). Beighley et al. (2003) also identified the overestimation of gauge records at
Gauge 11119940 during the 1995 January event. As for mean annual discharge, all three models
tend to overestimate for the study period, mainly due to the overestimation of subsurface flow
during dry seasons (Fig. 2d). This highlights challenges of simulating hydrologic processes in
semiarid regions under a Mediterranean climate.
Among the three hydrologic models, STP-HRR has the best overall performance (i.e.,
highest average NSE), mainly due to its better ability for capturing flood peaks than the other 2
models (Fig. 2, S2-S4). The peak performance is likely a result of the STP-HRR representing the
runoff generation process as an exponential relationship between soil moisture and runoff rates,
which makes runoff generation more sensitive to soil moisture dynamics as compared to the other
2 models. This algorithm is well suited to represent the significant nonlinearity of hydrologic
response to rainfall in the study region. RCM-HRR and VIC-HRR have similar overall
performance (i.e., similar average NSE), however, they represent hydrologic dynamics differently.
VIC-HRR tends to perform better in representing small peak flows than RCM-HRR while worse
in simulating mean flow (or total discharge volume) (Fig. 2, S2-S4). This is because as the wet
season proceeds, the lower soil layer is close to saturation (i.e., relative soil moisture is higher than
the threshold $W_s$ for VIC-HRR) which initiate the quadratic relationship between soil moisture and
subsurface runoff in VIC-HRR. This quadratic response to soil moisture conditions can lead to
much higher subsurface runoff (2-3 magnitudes higher than that of RCM-HRR), which contributes
to the lower performance in reproducing the total volume of discharge. This also explains that
VIC-HRR generates the highest subsurface runoff during the wet season (Fig. 3). In addition, VIC-
HRR also generates the most surface runoff during wet season (Fig. 3). This is because when soil
is almost saturated, surface runoff in VIC-HRR is almost a linear function of precipitation with a
coefficient of 1 (much larger than RCM-HRR which is 0.2 ($C_2$) and STP-HRR which is around
0.5 depending on the watershed topography).  The higher surface and subsurface runoff generated
by VIC-HRR leads to the overestimation of mean annual flow (Fig. 2d). However, there are no in-





situ measurement of surface and subsurface fluxes, and it is difficult to evaluate model
performance for these quantities individually or as a ratio. In Fig. 3, the simulated surface and
subsurface runoff from National Land Data Assimilation Systems VIC model (NLDAS-VIC)
output is also shown for purpose of comparison. A similar pattern, i.e., a very high subsurface
runoff, even higher than surface runoff, during wet season, can be found from NLDAS-VIC
simulations. The surface runoff of NLDAS-VIC is lower than those generated by the models in
this study, which is probably because of the difference in precipitation inputs. The NLDAS
precipitation input is lower during wet season than that used in this study for the study region. In
addition, the difference in spatial resolutions of precipitation (0.125° for NLDAS vs. 0.0625° for
this study) can also contribute to the difference in simulated runoff.
These results may suggest that STP-HRR is more suitable than VIC-HRR in representing
hydrologic processes in Mediterranean regions where 80% annual precipitation is concentrated in
a short period (roughly 3 months). As the wet season proceeds, the soil is close to saturation
conditions, under which the saturation excess overland flow is dominant. That explains why STP-
HRR performs best in this study region. VIC-HRR is probably more suitable to the regions where
precipitation events are sparsely distributed where soil is not easy saturated. Although RCM is an
empirical method, it performs fairly well in this study, mainly because it captures the nonlinearity
of hydrologic processes through a switch between dry and wet surface runoff coefficients ($C_1$ and
$C_2$) based on the soil moisture conditions.
Three sets of parameters with the best performance (assessed by NSE) were selected for
each model (Fig. 4). For most of parameters, the selected optimal values are very close, except $C_1$
and $K_{s\_all}$ in RCM-HRR, suggesting that most parameters are important factors controlling model
performance. For some parameters whose optimal values are close to their range boundaries, for
example, $K_{ss\_all}$, $W_s$, $\varphi_{sat}$ and $Q_m$, wider and physically acceptable ranges were tested and similar
results were obtained, which suggests the ranges of these parameters defined in this study (Table
1) are reasonable.
3.2 Uncertainty analysis
For the 28 major watersheds in SBC, the projected changes in $Q_m$ during 2081-2100 as
compared to historical period 1986-2005, range from -80% to 150% (Fig. 5). The median changes
for each of these major watersheds are slightly above 0%, varying between 1% and 10%. The



major uncertainty sources are GCM and RCP, which account for about 50% of the total
uncertainty. Among the first order factors (i.e., GCM, RCP, hydrologic model and
parameterization), hydrologic model ranks third after GCM and RCP, accounting for about 10-
20% of total uncertainty. In contrast, parameterization only introduces less than 1% of the total
uncertainty. The remaining 30-40% uncertainty is from the second, third and fourth order
interactions between the four major sources. The projected relative changes in $Q_P$ and $Q_{100}$ during
2081-2100 compared to 1986-2005 are similar in magnitudes, both varying from -90% to 240%
(Fig. 6 and Fig. 7). The median changes in $Q_P$ and $Q_{100}$ for each watershed are higher than those
of $Q_m$, ranging between 10-40%. For most of watersheds, GCM and RCP are the two major
uncertainty contributors for $Q_P$ and $Q_{100}$, accounting for 40-60% of total uncertainties. Hydrologic
model contributes about 10-30% of total uncertainties in $Q_P$ and $Q_{100}$. Compared to $Q_m$, $Q_P$ and
$Q_{100}$ get more uncertainty from the hydrologic models, which is likely due to highly nonlinear
rainfall-runoff behavior and larger differences between runoff generation methods in generating
peak flows as compared to average flow conditions.

Changes in $Q_m$, $Q_P$ and $Q_{100}$ are higher under RCP 8.5, but the uncertainties are also higher

(Fig. 8), which suggests the uncertainties from RCPs are mainly introduced by RCP 8.5. In Mission
Creek watershed (USGS gauge No. 11119750), the probability of increase in Qm under RCP 4.5
is only 51%. However, this probability increases to 64% under RCP 8.5. For the less frequent
events ($Q_P$ and $Q_{100}$), the probabilities of positive changes are higher: 78% and 85% for $Q_P$ and
$Q_{100}$, respectively, under RCP 8.5. This implies that if RCP 8.5 happens in the future, the extreme
events will probably get intensified.

Compared to previous studies (e.g., Vetter et al. (2015), Schewe et al. (2014),  Hagemann

et al. (2013);(Troin et al., 2018), and Asadieh and Krakauer (2017)), this work identifies relatively
lower uncertainty contributions from hydrologic models. This is mainly because in this study the
models use the same model configuration including the model unit definition (irregular
catchments) and the hillslope routing scheme ("open-book" assumption), which reduces the
difference between hydrologic models. Here, a common calibration approach is also used to
eliminate user/method bias which is common in studies that consider more than one hydrologic
model. In contrast, the hydrologic models used in previous studies are the individual models which
use their own model configurations. For example, the VIC model (here VIC refers to the original
VIC models, and is different from the model used in this study; to clarify, in following text, VIC



refers to the original VIC model while VIC-HRR refers to the model used in this study) uses the
grid-based model units ignoring the spatial arrangement and has its own routing scheme which
adopts the synthetic unit hydrograph concept. These differences between models probably resulted
in the larger uncertainties in the simulation from hydrologic models in previous studies.

Different from previous studies, the hydrologic model uncertainty in this study only comes

from the runoff generation algorithms. This can provide useful information for selecting
hydrologic models for climate change impact analysis. The results in this study imply that selecting
an appropriate runoff generation algorithm suitable to the regions of interest and the study targets
(e.g., total volume or extremes) can reduce uncertainties by 10-30%, especially for the extreme
quantities (e.g., 100-yr flood discharge). Compared to the runoff generation algorithms, model
parameterization plays a negligible role (less than 1%) in the total streamflow uncertainty. This
suggests that the parameter equifinality (or non-uniqueness) is less of a concern when quantifying
climate change impacts on hydrologic fluxes using an ensemble of GCM forcings. In this study,
only one routing scheme is investigated. Although there are fewer variants of routing algorithms
as compared to runoff generation methods, the choice of different routing methods can still make
a difference in the total uncertainties in streamflow simulation, especially when the model
configurations are different, for example, the routing schemes in VIC model and HRR model.
Therefore, further study integrating different routing algorithms should be conducted to evaluate
the uncertainties in simulated streamflow resulted from both process models (runoff generation
model and routing model), which can be useful to guide stakeholders to select appropriate
hydrologic algorithms for climate change impacts analysis and develop actionable adaptation and
mitigation strategies.
**4 Conclusions**

A modeling framework which integrates multiple runoff generation algorithms (VIC, STP

and RCM) with the Hillslope River Routing model (HRR) is developed. Forced with an ensemble
of GCM projections under different emission scenarios, this framework is able to quantify the
climate change impacts on streamflow and evaluate the associated uncertainties from different
sources (i.e., RCPs, GCMs, hydrologic process models and parameterization). The results in this
study show that the median changes in mean annual discharge for the major watersheds in SBC
are 1-10%, with an uncertainty of 230% (-80% to +150%); the median changes in annual peak





discharge and 100-yr flood discharge are higher than those of mean annual discharge, varying
between 10% and 40%, but with a higher uncertainty of 330% (-90% to +240%). For these
uncertainties, GCM and RCP are the first two major contributors, accounting for more than 50%
of total uncertainties at most watersheds in SBC, while hydrologic process models (i.e., runoff
generation modules) contribute between 10% and 30% among watersheds with the remaining 20-
40% of the uncertainty coming from the interactions between these individual sources. Hydrologic
model parameters alone contribute less than 1% of the uncertainty, which suggests the parameter
equifinality should not be a concern when analyzing climate change impacts using ensembles of
climate models projections. The results based on the BMA analysis indicate that there is a high
probability (up to 85%) that streamflow, especially the extreme quantities like $Q_{100}$ under RCP
8.5, will increase in SBC.

Unique to this framework, the uncertainties from different hydrologic model components

(e.g., runoff generation process) and associated model parameterizations can be identified and
quantified. In this study, only one routing scheme is integrated, however, this framework is capable
of incorporating multiple routing methods to quantify their contributions to the uncertainties in
simulated hydrologic variables. These information can help stakeholders with different focuses
(e.g., water resources, risk controls or ecosystem conservation) customize and optimize their
selections of hydrologic models and make actionable adaptation decisions under the changing
climate.
**Code availability**

504        The source code supporting this work is available on Github:

https://github.com/dongmeifeng-2019/HydroUncertainty
**Author contribution**
D. Feng and E. Beighley designed the experiments and D. Feng developed the model code and
performed the simulations. D. Feng and E. Beighley prepared the manuscript.
**Competing interests**
The authors declare that they have no conflict of interest.



**Acknowledgments**

This research was supported by the Santa Barbara Area Coastal Ecosystem Vulnerability Assessment (SBA CEVA) with funding from the NOAA Climate Program Office Coastal and Ocean Climate Applications (COCA) and Sea Grant Community Climate Adaptation Initiative (CCAI), and the National Science Foundation's Long-Term Ecological Research (LTER) program (Santa Barbara Coastal LTER - OCE9982105, OCE-0620276 and OCE-123277). The authors thank Dr. David Hadka at Pennsylvania State University and Chinedum Eluwa at University of Massachusetts, Amherst, for their help with setting up the Borg MOEA.



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




**Table 1**: Calibrated parameters for all 3 models

| Parameters | Description | Unit | Range | RCM-HRR | VIC-HRR | STP-HRR |
|---|---|---|---|---|---|---|
| $K_{s\_all}$ | coefficient to adjust surface roughness | - | 1-20 | ✓ | ✓ | ✓ |
| $K_{ss\_all}$ | coefficient to adjust horizontal hydraulic conductivity | - | 10-200 | ✓ | ✓ | ✓ |
| $K_{sat\_all}$ | coefficient to adjust vertical hydraulic conductivity | - | 0.01-5.0 | ✓ | | |
| $C_1$ | dry runoff coefficient | - | 0-0.3 | ✓ | | |
| $C_2$ | wet runoff coefficient | - | 0.2-0.8 | ✓ | | |
| $\theta_t$ | soil moisture threshold separating dry and wet conditions | - | 0.2-0.8 | ✓ | | |
| $b_{in}$ | Infiltration curve shape parameter | - | 0.005-0.5 | | ✓ | |
| $D_m$ | maximum baseflow | $m\cdot d^{-1}$ | 0 -0.037 | | ✓ | |
| $D_s$ | fraction of $D_M$ where non-linear baseflow begins | - | 0 -0.005 | | ✓ | |
| $W_s$ | fraction of the maximum soil moisture where non-linear baseflow occurs | - | 0.92-1.0 | | ✓ | |
| $f_{over}$ | Surface runoff coefficient | $m^{-1}$ | 0.1-5 | | | ✓ |
| $f_{drain}$ | Subsurface runoff coefficient | $m^{-1}$ | 0.1-5 | | | ✓ |
| $Q_m$ | maximum baseflow | $m\cdot d^{-1}$ | 0.864-1728 | | | ✓ |
| $\varphi_{sat}$ | Saturated suction head in the soil | m | -3.05–0 | | | ✓ |



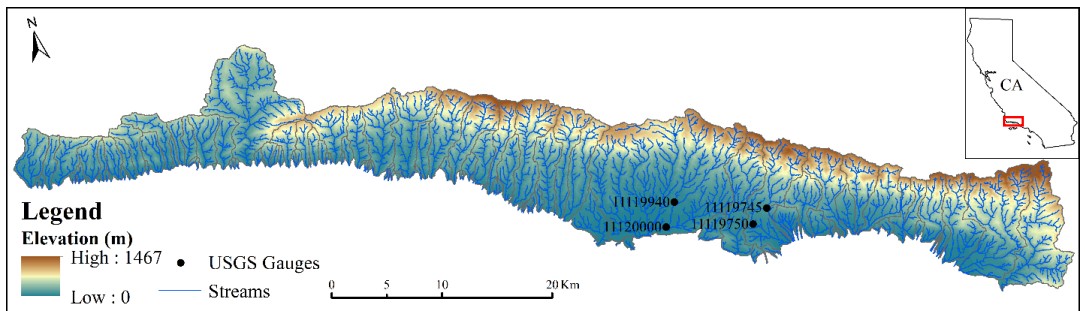

**Figure 1: Study region with USGS streamflow gauges. The inset figure indicates the location of SBC within the state of California (CA).**



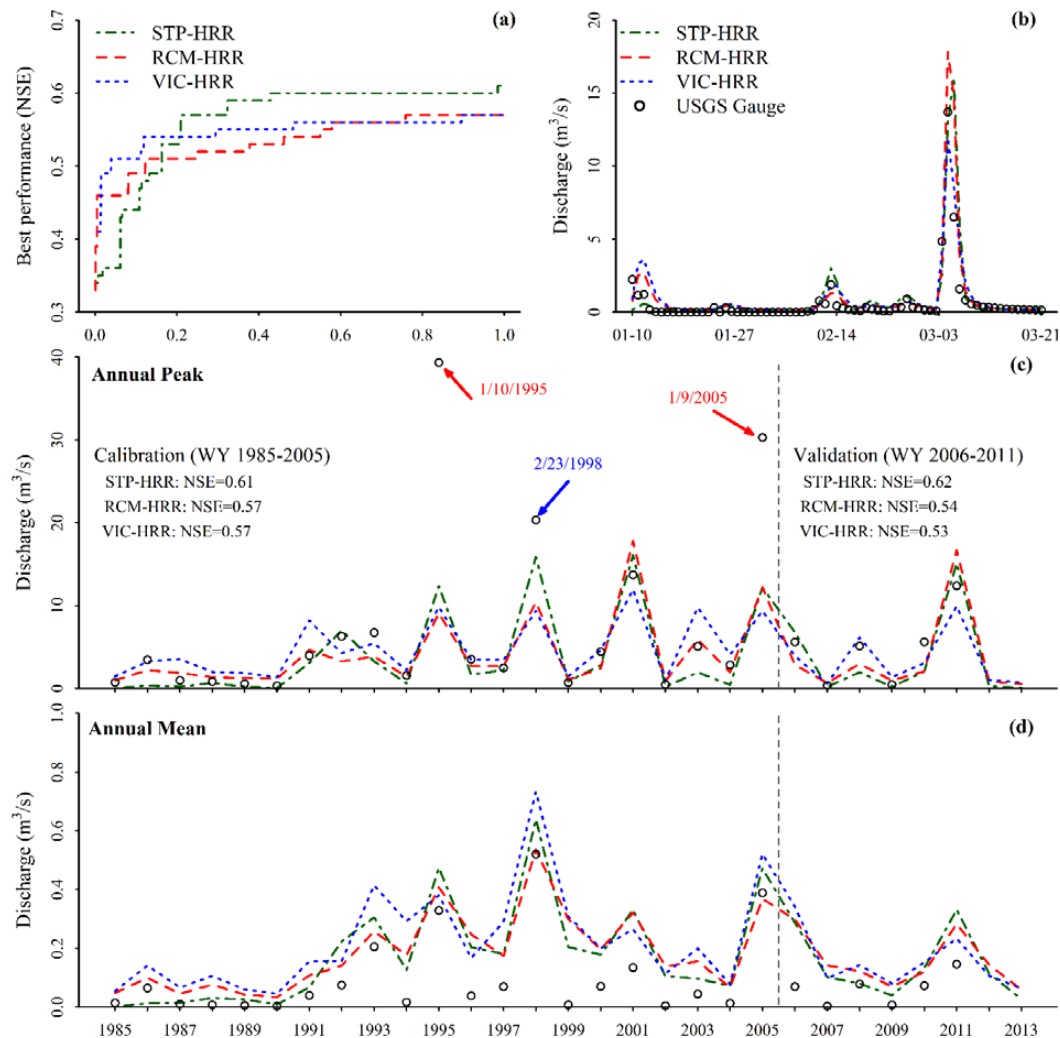

**Figure 2 Model performance for calibration and validation periods: (a) model performance**
**(assessed by NSE) during calibration process, x axis is the normalized calibration process; (b)**
**hydrographs simulated by 3 calibrated models and measured by USGS gauge; in order to show the**
**details of the hydrographs, they are zoomed in to the wet season in 2001; the model performance is**
**similar in other years; (c) simulated annual peak flow during calibration (water year 1985-2005)**
**and validation (water year 2006-2011) periods as compared with in situ observations; texts indicate**
**model performance (i.e., NSE) in reproducing historical hydrographs for both periods; the points**
**highlighted in red arrows indicate the events were not reproduced by models due to the input (e.g.,**
**precipitation or discharge observation) bias; the point highlighted in blue arrow is similar to those**
**in red but at a lower probability; and (d) simulated and observed annual mean flow during**
**calibration and validation periods. For clarity, only results for Mission Creek watershed (USGS**
**gauge NO. 11119750) are shown here; results for other gauged watersheds are similar and can be**
**found in the Supporting Information (Figure S1-S3).**



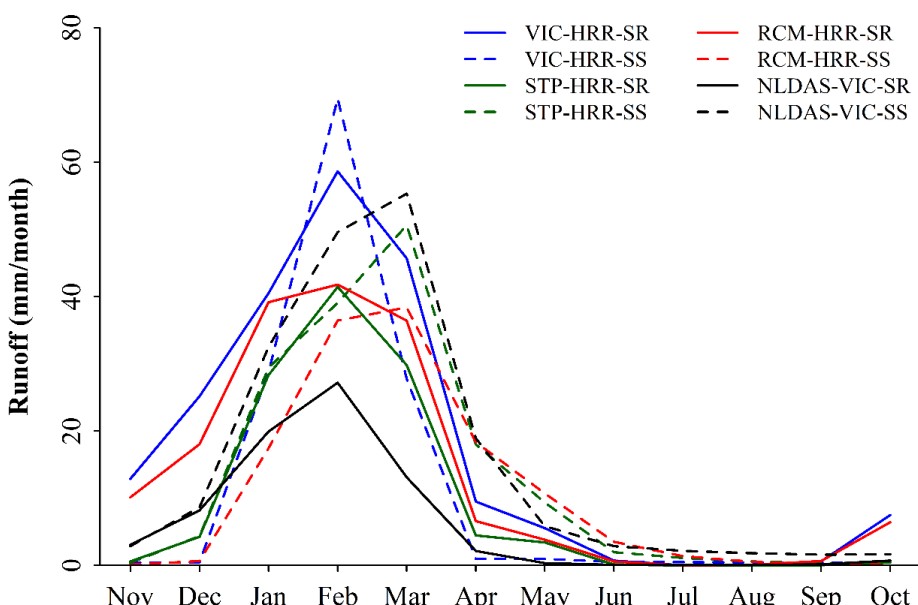

**Figure 3: Simulated monthly surface and subsurface runoff from Mission Creek watershed (USGS gauge NO. 11119750) by three models for the calibration period (water year 1985-2005). Surface runoff is denoted by 'SR' and subsurface runoff is denoted by 'SS' in this figure. Monthly surface and subsurface runoff from National Land Data Assimilation Systems (NLDAS) VIC model simulation for the same period are shown here for comparison purpose.**

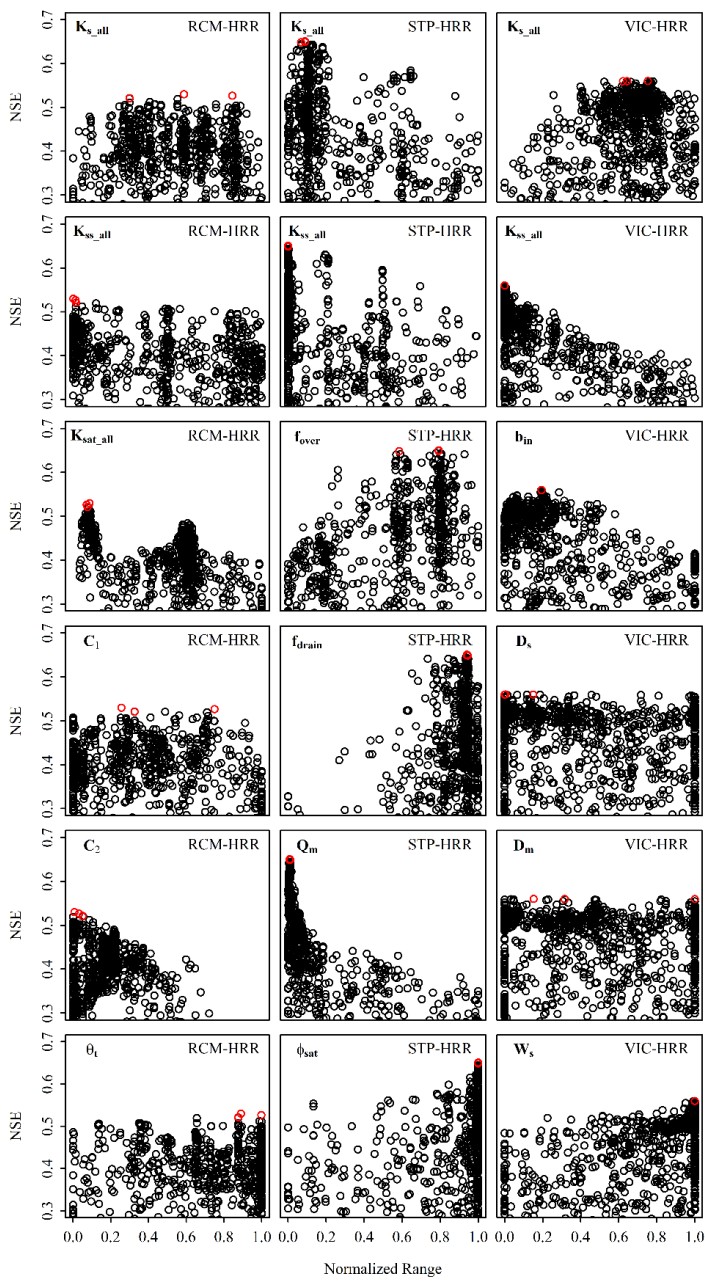

**Figure 4. Parameters sampled during calibration process and their corresponding performance**
**(assessed by NSE). The black circles are parameter samples within the predefined ranges (shown in**
**Table 1) and the red circles indicate the optimal values used for further uncertainty analysis. The**
**parameter values are normalized by their ranges, so the range of x axis in all plots is 0-1. The**
**parameters were sampled throughout their whole ranges, however, for clarity, samples with NSE**
**lower than 0.3 are not shown in this figure.**

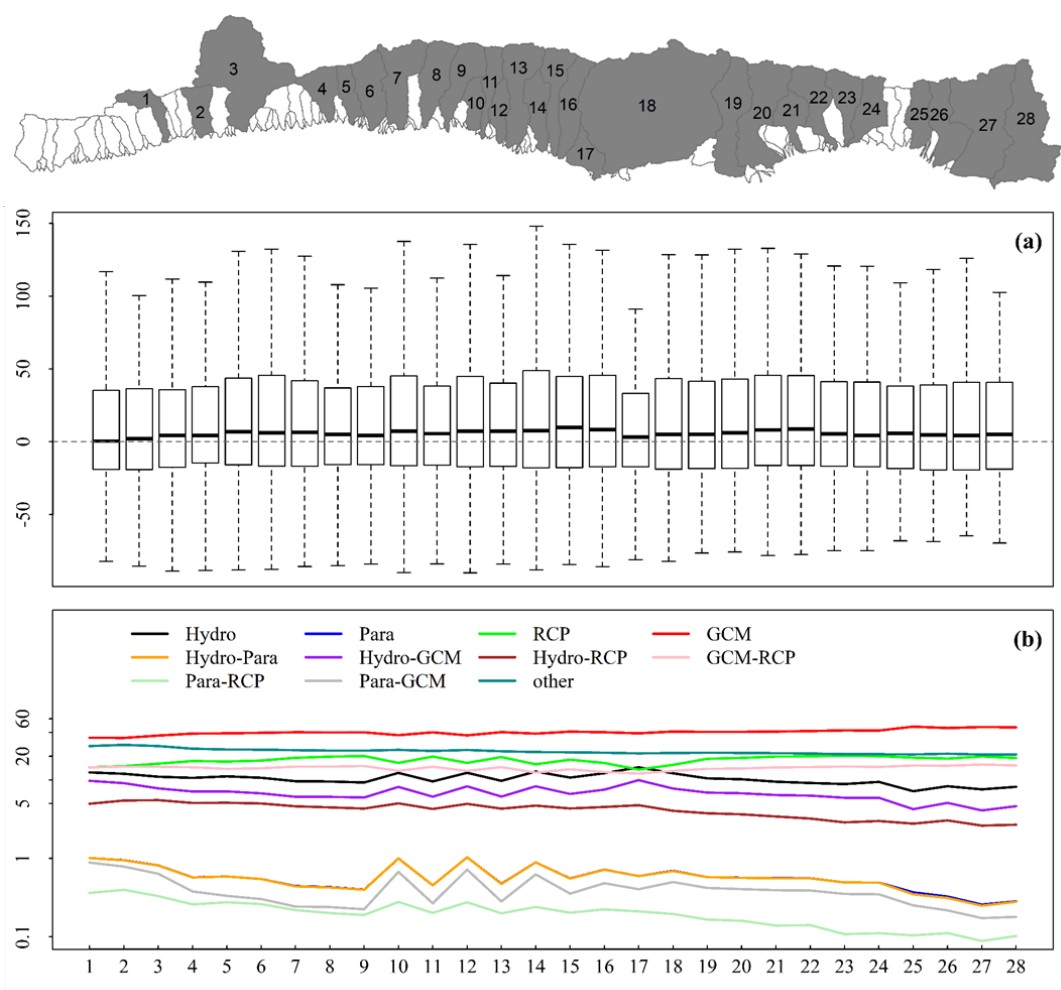

**Figure 5. (a) Projected relative changes (%) in annual mean discharge ($Q_m$) in the major SBC watersheds (indicated by the grey watersheds in the map) during 2081-2100 as compared to historical period (1986-2005); each bar depicts relative changes in minimum, maximum, median, 1st and 3rd quartiles for the ensemble outputs; bars from left to right spatially corresponding to watersheds from west to east. For clarity, only watersheds with drainage areas larger than 7 km$^2$, which account for roughly 83% of the study area, are shown. (b) Relative sources (%) of the uncertainties in the projected changes at each of these watersheds; the category "other" is the uncertainty from the 3rd and 4th orders of interactions between the 4 major sources (i.e., GCMs, RCPs, Hydrologic models, denoted by "Hydro" and parameters denoted by "Para")**

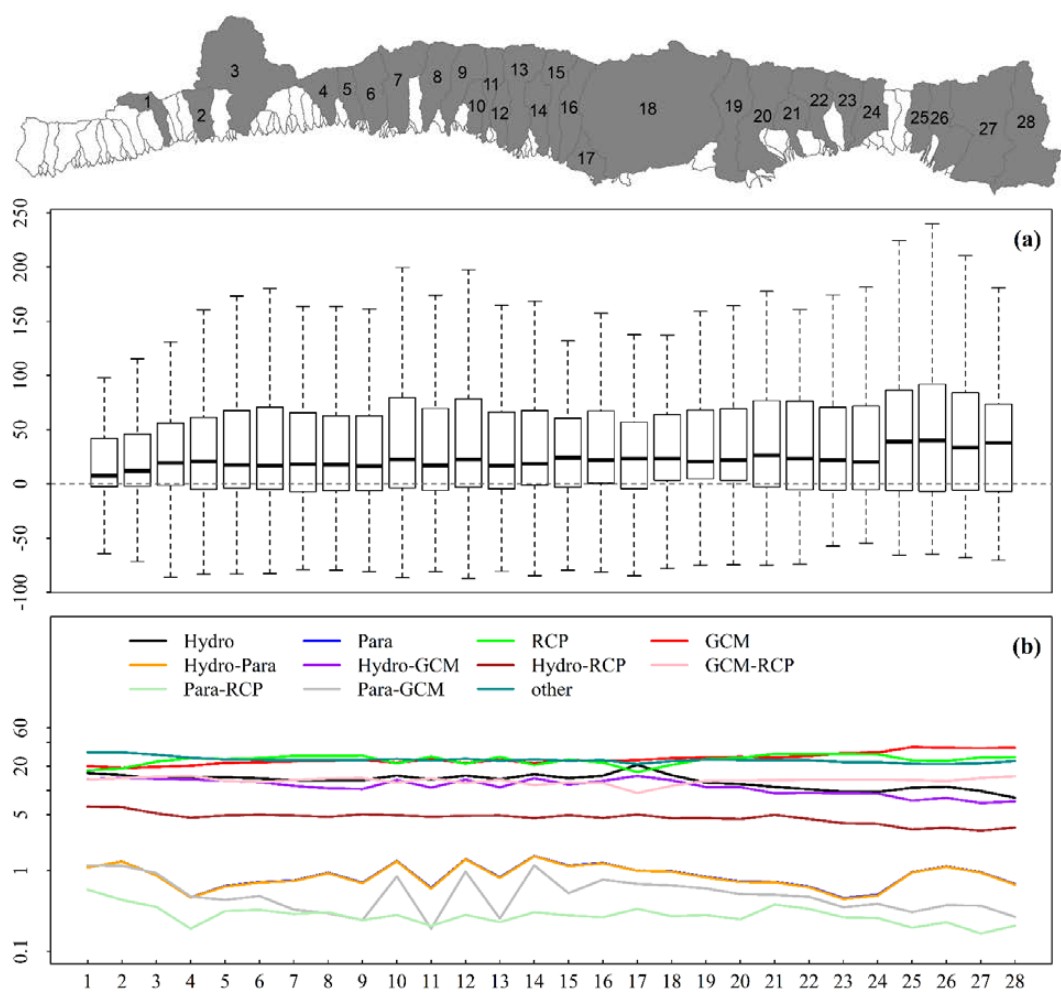

**Figure 6**. **(a) Projected relative changes (%) in annual mean discharge ($Q_p$) in the major SBC watersheds (indicated by the grey watersheds in the map) during 2081-2100 as compared to historical period (1986-2005); each bar depicts relative changes in minimum, maximum, median, 1st and 3rd quartiles for the ensemble outputs; bars from left to right spatially corresponding to watersheds from west to east. For clarity, only watersheds with drainage areas larger than 7 km$^2$, which account for roughly 83% of the study area, are shown. (b) Relative sources (%) of the uncertainties in the projected changes at each of these watersheds; the category "other" is the uncertainty from the 3rd and 4th orders of interactions between the 4 major sources (i.e., GCMs, RCPs, Hydrologic models, denoted by "Hydro" and parameters denoted by "Para")**

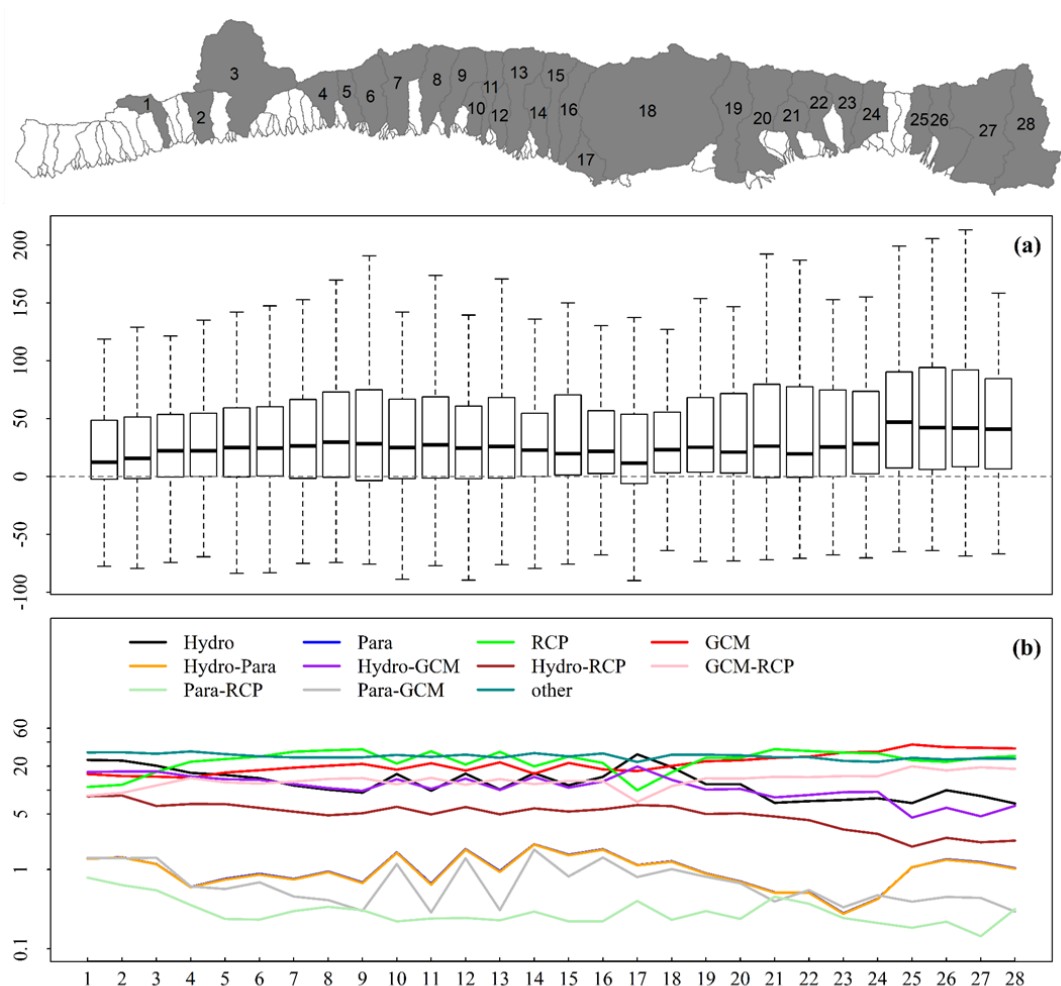

**Figure 7. (a) Projected relative changes (%) in 100-yr flood discharge ($Q_{100}$) in the major SBC watersheds (indicated by the grey watersheds in the map) during 2081-2100 as compared to historical period (1986-2005); each bar depicts relative changes in minimum, maximum, median, 1st and 3rd quartiles for the ensemble outputs; bars from left to right spatially corresponding to watersheds from west to east. For clarity, only watersheds with drainage areas larger than 7 km², which account for roughly 83% of the study area, are shown. (b) Relative sources (%) of the uncertainties in the projected changes at each of these watersheds; the category "other" is the uncertainty from the 3rd and 4th orders of interactions between the 4 major sources (i.e., GCMs, RCPs, Hydrologic models, denoted by "Hydro" and parameters denoted by "Para")**

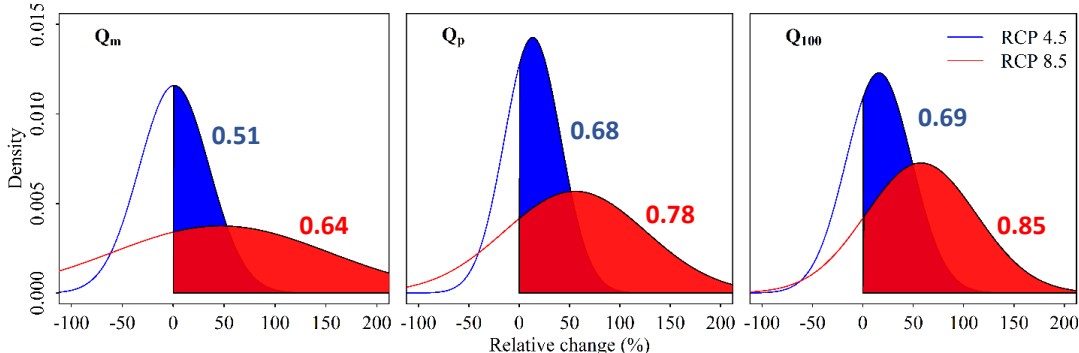

**Figure 8. Probability of changes in $Q_m$, $Q_p$ and $Q_{100}$ at Mission Creek watershed (No. 20 in Figure 5 map). The numbers in the plot are the probabilities of positive changes in $Q_m$, $Q_p$ and $Q_{100}$ (areas of shaded regions) under each emission scenario (blue numbers are for RCM 4.5 and red numbers are for RCP 8.5).**