# Peer review of "Identifying uncertainties in simulated streamflow from hydrologic model components for climate change impact assessments"

_Hydrology and Earth System Sciences, 2019_

## Referee Comment (RC1) · Anonymous Referee #1 · 7 Aug 2019

General Comments (overall quality of paper):

Overall, I think the paper was well written as a climate impacts assessment and application of uncertainty methods provided elsewhere. As stated in the Introduction, the goals of the paper were threefold:

1) Compare different hydrologic models 2) Quantify uncertainty associated with different modeling choices 3) Provide suggestions for studies looking at impacts of climate change

[Figure]

The paper accomplished these three goals (aside for one point which I address in the paragraph below). However, the authors don't make it clear how the field is moved forward even if all three goals are achieved. The authors' study appears to be similar to the Vetter et al (2015) study but in a different, and more homogeneous, domain. As-is, the authors conducted a very detailed assessment of climate change impacts on streamflow in Santa Barbara County. Their assignment of uncertainty to different modeling components followed methods similar to those in previous studies like Addor et al (2014), Vetter et al (2015), Hattermann et al (2018), Chegwidden et al (2019). I did not see any truly novel methods proposed, thus making the paper seem more like an, albeit very rigorous, report. As is, the study is appropriate for a climate impacts assessment journal, but the findings are insufficiently new to warrant publication in HESS.

To make the manuscript more relevant to HESS, I suggest a handful of other potential additions to deepen the analysis. Would it be possible to expand the analyses conducted here to other domains and thus do an intercomparison across different regions? For example, the findings in Figures 5 through 7 are relatively uniform across region and metric. Perhaps the authors could probe deeper into those comparisons by conducting more analyses in other regions or with other metrics? By expanding the analysis to other regions and metrics the study could test how sensitive the uncertainty analysis is to the research question of interest.

Another potential avenue of analysis could be a deeper understanding of the parameter space. I am skeptical about the finding that parameterizations explained little of the uncertainty since it appeared (from Figure 4) that the values within the different parameter sets evaluated were actually quite similar. Since it appears that you have those parameter sets available, would it be possible to expand the analysis to include more parameter sets? That could buy more confidence in the current analyses.

References not included in the current manuscript:

Addor, N., Rössler, O., Köplin, N., Huss, M., Weingartner, R., & Seibert, J. (2014). Robust changes and sources of uncertainty in the projected hydrological regimes of Swiss catchments. Water Resources Research, 50(10), 7541–7562. https://doi.org/10.1002/2014WR015549

Chegwidden, O. S., Nijssen, B., Rupp,D. E., Arnold, J. R., Clark, M. P.,Hamman, J. J., et al. (2019). How do modeling decisions affect the spread among hydrologic climate change projections? Exploring a large ensemble of simulations across a diversity of hydroclimates. Earth's Future,7,623–637. https://doi.org/10.1029/2018EF001047

Hattermann, F. F., Vetter, T., Breuer, L., Su, B., Daggupati, P., Donnelly, C., . . . Krysnaova, V. (2018). Sources of uncertainty in hydrological climate impact assessment: A cross-scale study. Environmental Research Letters, 13(1). https://doi.org/10.1088/1748-9326/aa9938

=================================

Specific Comments (individual scientific questions/issues):

L14- In the abstract, you mention that identification and uncertainties are rarely studied. This is not true. It is increasingly common (see, for example, the three references above).

L139-141 – Is the monthly 1 degree aerosol optical depth product sufficient for calculating radiation at the scale you are working at?

L154-156 – Should there be some discussion about the fact that, regardless of subbasin size (which, as the authors state, ranges between 0.1 and 135 kmˆ2) the parameters are averaged across each subbasin?

L175-254 – I'm not sure the specificity is necessary for each of the hydrologic models in the main text. I would suggest moving the conceptual plot from the supplemental text to the main text and moving the mathematical explanations to the supplemental text. This would save you space in the main body of the text, improving readability,

while letting your story come through easier. With that space you could fill in with more details on the calibration methods.

L260-263 – The definitions of Kss and Ks would probably best fit in the description of the routing model since they are from that model.

L268-269 – How are the three different optimal parameter sets selected? Are they very different parameter sets? As in, are they likely to be in very different parts of the overall calibration space/range of parameter values? Or are they likely to be relatively similar? If they are similar, does that explain the less than 1% uncertainty explained by parameterization referenced in L426 in the results? I see in Figure 4 that some of the parameter values for some models are indeed quite similar (e.g. Kss_all for RCM-HRR). How does this affect your conclusions about the minimal contribution of parameterization toward total uncertainty?

L320-324 – Do the authors conduct their performance weighting based upon the GCM simulations? Or do they do it using the historical meteorological forcing data (in this case Livneh et al)? The latter would be appropriate, since the former would not match the actual weather experienced by the region.

L360-363 – Is the climate data, even though it was downscaled to the 1/16th degree scale, appropriate for the scale of the subbasins the authors are evaluating (for instance the basin that is only 0.1 km^2)? As the authors suggest in L360-363, they note substantial biases in the precipitation that, in one example case, doesn't even provide enough water to account for streamflow in absence of ET. Did the authors modify precipitation at all to account for this? If not, do the authors think that some other modification of the precipitation forcing would be appropriate? Also, in Figure 2 caption: (a) what does "normalized calibration process" mean?

L437-438 – "Changes in Qm, Qp and Q100 are higher under RCP 8.5, but the uncertainties are also higher (Fig. 8), which suggests the uncertainties from RCPs are mainly introduced by RCP 8.5." Could you clarify this statement? I think there may

be some conflation in the sources of the uncertainties in this argument. In looking at Figure 8, we see that the distributions are very different between RCP45 and RCP85. However, in your ANOVA formulation, the comparison of the different model choices really just looks at the differences in the means. Thus, attributing the uncertainty to RCP 8.5 can't be made by these figures alone, since you are only comparing two choices. If you are referring to the large standard deviation of the RCP85 distribution, then that uncertainty contribution would actually be a higher-order interaction of RCP and something else (perhaps GCM?).

L394-396 – I assume the reference to the National Land Data Assimilation System VIC model set-up is the one referenced at the following DOI? (https://doi.org/10.5067/ELBDAPAKNGJ9) If so, it needs a citation and perhaps some explanation as to why this is used as a suitable comparison.

L446-449/456-457 – How can you justify that model configurations (e.g. irregular catchments or routing schemes) are the reason that hydrologic models played a smaller role in your uncertainty findings?

L449-451 – What do the authors mean by "a common calibration approach is also used to eliminate user/method bias"?

L461-462 – Is reducing the uncertainty the goal for an impacts assessment? Would not the goal actually be to reveal the uncertainty present, and thus actually focus on multiple hydrologic models as the authors suggest that their selection accounts for a sizeable portion of the uncertainty space?

L471-475 – At the relatively small scale which you are working, how is routing impactful?

L483 – How do you define uncertainty of 230%? Is that the range? Or +/- 2 standard deviations?

================================= Technical Corrections (typing errors,

etc):

L43 – "cause" not "causes"

L69-70, L81 – Confusing sentences/phrasing

L220 – "matric" not "metric" – there are many other language typos (e.g. L222 "expresses" should be "expressed") sprinkled throughout the text, but I imagine that with another read-through these issues could be resolved.

Overall, there are small language errors throughout the manuscript which the vast majority of the time don't interfere with understanding but are somewhat distracting. A careful reading would help resolve these.

---

## Referee Comment (RC2) · Anonymous Referee #2 · 21 Dec 2019

This manuscript tries to investigate the uncertainties resulting from different hydrological model components when assessing the impacts of climate change on streamflow. To do so, they design a modeling framework that incorporates three runoff yield schemes, one runoff routing scheme, several GCM and RCP. I think the topic is interesting and the manuscript is overall well-prepared. However, I think there are still several issues have to be addressed before considering for publication in HESS.

(1) The authors choose annual mean discharge, annual peak discharge or 100-yr flood discharge to analyze the uncertainties. I doubt if it's meaningful to investigate annual mean values in a 750 km$^2$ catchment. In figure 5 they even investigate changes and uncertainties in much smaller sub-basin. Because I think, according to their methodology, in such a small catchment the annual mean runoff is simply controlled by precipitation and evaporation. On the other hand, when investigate the annual peak values (here it's not clear how they define 'peak' values, from daily or hourly?), the routing may play a more significant role in the timing and magnitude of simulated streamflow. My concern is if the authors can still reach the same conclusions if they use daily streamflow when perform uncertainties analysis because I believe in such small catchment different runoff yield schemes have more effects on daily streamflow instead annual streamflow.

(2) I also want to hear opinions from the authors regarding the choose of runoff yield scheme. When perform regional or global simulations using LSM, people usually can only use one runoff yield option, either saturation-excess (e.g. NoahMP, CLM) or infiltration-excess (e.g. VIC). However, when focus on the specific catchment, you can definitely choose a runoff yield scheme that is suitable for the hydrological regime of that catchment. I'm not challenging your work, just want to hear some discussion.

(3) Line 134-141. The authors use MODIS products to estimate the PET, However, they don't provide any detail regarding how to convert PET into ET for runoff yield simulation. In eq(1)~(7) I don't see any variable related to ET.

(4) Line 255. The authors calibrate several parameters related to runoff. But they don't document how they fix the value of soil depth, from dataset or by calibration. In Line 233 they state that the soil depth is based on a previous study but I don't see any description in (Feng et al., 2019). In their modeling framework, they use quite simple water balance scheme to account for the soil water movement, in this case the soil depth is an important variable determining the soil water holding capacity.

(5) Line 256. sim-topmodel uses groundwater depth to calculate runoff yield. Do you spin up the model to reach the equilibrium state?

(6) Line 290. If I understand correctly, here should be "parameter", which is different from "parameterization"

---

## Short Comment (SC1) · 30 Dec 2019

This paper presents some limited results of evaluating the impact of different formulations in runoff generation schemes when simulating streamflow. My major objection with the paper is that it really is not assessing the uncertainty but rather the variability of the simulated streamflow and how each of the forcings, model parameters or formulations contribute to it. Although that is valuable in itself, the authors claim that the objective is to identify the uncertainties in the context of climate change simulations. However, that is not what was done here. The calibration of the parameters was done
using an observation-based forcing dataset and although I can understand the rationale, I believe that any calibration of parameters should have been done in a way that would emulate the intended application (i.e. using GCM output to drive the hydrology model). I believe historical simulation are available from CMIP5 and if so they should be used to evaluate the actual uncertainty of simulated streamflow within the framework that the authors have developed. The end of 21st century simulations should be a final experiment (if included at all) given the objective of the paper. Consequently, I recommend major revisions before publication that will include new simulations that test the different model parameter sets, runoff generation schemes and downscaled GCM output for the period when streamflow measurements are available, so that the actual uncertainty can be quantified. In addition, I believe the study area is rather limited and an opportunity is being missed by not including additional basins with different physiography and climate. Some additional comments are outlined below:

* How does the uncertainties in the prescribed ET affect the results? Why weren't they accounted for? * Abstract needs some attention, especially after l. 21 in terms of cohesiveness. Right now, it reads as bullet points stitched together. * Some proofreading needed for redundant articles and grammatical errors. * l. 53: what is the need for naming the "land-atmosphere interactions" as "runoff generation process" when the latter is clearly one of the processes that manifest from those interactions? * l. 175- : Not sure whether this much detail is needed for the description of the runoff generation models, since they are well established. * l. 354: does that mean that there is bias in the validation data (i.e. streamflow)? * l. 362-363: this highlights another problem that has not been addressed in this study: the downscaling of GCM outputs to drive the hydrology model.

[Figure]

---

## Author Comment (AC2) · 5 Jan 2020

**Reviewer #2**

This manuscript tries to investigate the uncertainties resulting from different hydrological model components when assessing the impacts of climate change on streamflow. To do so, they design a modeling framework that incorporates three runoff yield schemes, one runoff routing scheme, several GCM and RCP. I think the topic is interesting and the manuscript is overall well-prepared. However, I think there are still several issues have to be addressed before considering for publication in HESS.

The authors choose annual mean discharge, annual peak discharge or 100-yr flood discharge to analyze the uncertainties. I doubt if it's meaningful to investigate annual mean values in a 750 km2 catchment. In figure 5 they even investigate changes and uncertainties in much smaller sub-basin. Because I think, according to their methodology, in such a small catchment the annual mean runoff is simply controlled by precipitation and evaporation. On the other hand, when investigate the annual peak values (here it's not clear how they define 'peak' values, from daily or hourly?), the routing may play a more significant role in the timing and magnitude of simulated streamflow. My concern is if the authors can still reach the same conclusions if they use daily streamflow when perform uncertainties analysis because I believe in such small catchment different runoff yield schemes have more effects on daily streamflow instead annual streamflow.

**Response:** The annual mean discharge was defined as the average of daily streamflow in a year. To clarify it, we have inserted the following sentence in the manuscript:
*"Here, the annual mean discharge was defined as the average of daily streamflow in a year."*
The annual peak discharge was defined as the maximum daily streamflow in a year. It was described in L322-323: *"annual maximum daily discharge ($Q_p$)"*

I also want to hear opinions from the authors regarding the choose of runoff yield scheme. When perform regional or global simulations using LSM, people usually can only use one runoff yield option, either saturation-excess (e.g. NoahMP, CLM) or infiltration-excess (e.g. VIC). However, when focus on the specific catchment, you can definitely choose a runoff yield scheme that is suitable for the hydrological regime of that catchment. I'm not challenging your work, just want to hear some discussion.

**Response:** This is a good point. One of the main objectives of this study was to *"evaluate and compare the performance of hydrologic models with different approaches representing runoff generation process…"* (L111-113). The results in this study showed that STP performs better than the other two methods. This finding can be informative for future studies associated with hydrologic model selection. We have inserted the following discussion in the manuscript:

*"This study can also provide useful information for selecting hydrologic models for climate change impact analysis. As discussed in section 3.1, the STP-HRR model is more suitable than the other two models for the study region, mainly due to its ability to represent the non-linear hydrological response to precipitation forcings. This implies hydrologic models adopting the saturation excess runoff generation algorithms may be more suitable for areas with a Mediterranean climate. The uncertainties from hydrologic models are larger than those from the hydrologic model parameters for all hydrologic variables (e.g., discharge, runoff and seasonality), suggesting the inter-model variability is larger than the intra-model variability*

*(from model parameters). This implies that model selection is more important than the parameter selection, and that the parameter equifinality (or non-uniqueness) is less of a concern when quantifying climate change impacts on hydrologic fluxes when using an ensemble of GCM forcings. In this study, only the runoff generation algorithm was investigated. Other hydrologic model components, such as ET algorithm and routing method, also have many variants. The choice of these components can also make a difference in the total uncertainties in simulated runoff and streamflow. Therefore, further study integrating different algorithms for these components can be conducted in the future. This complete analysis can be useful to guide stakeholders to select appropriate hydrologic algorithms for climate change impacts analysis and to develop actionable adaptation and mitigation strategies.*"

Line 134-141. The authors use MODIS products to estimate the PET, However, they don't provide any detail regarding how to convert PET into ET for runoff yield simulation. In eq(1)~(7) I don't see any variable related to ET.

**Response:** The ET was extracted from soil at each time step, and then the soil content was updated which was used for water balance calculation in next time step. We have added the following content to the manuscript:

*"The evapotranspiration was estimated using Eq. S15…. After the water fluxes (runoff, ET and water movement between soil layers) were determined, the soil moisture was updated which would be used for the water balance calculation in the next time step."*

$$ET = min\ (PET, W - W_{min}) \hspace{2cm} (S15)$$

*where PET is the potential evapotranspiration estimated using the method proposed by Raoufi and Beighley (2017); W is water content in the upper soil layer; $W_{min}$ is the minimum water content in the soil, defined as $0.15 \times W_s$; $W_s$ is soil water content as saturation.*

Line 255. The authors calibrate several parameters related to runoff. But they don't document how they fix the value of soil depth, from dataset or by calibration. In Line 233 they state that the soil depth is based on a previous study but I don't see any description in (Feng et al., 2019). In their modeling framework, they use quite simple water balance scheme to account for the soil water movement, in this case the soil depth is an important variable determining the soil water holding capacity.

**Response:** The soil depth data was originally from the Soil Survey Geographic (SSURGO) Data Base for Santa Barbara County. This reference has been added to the manuscript.

Line 256. sim-topmodel uses groundwater depth to calculate runoff yield. Do you spin up the model to reach the equilibrium state?

**Response:** Yes. We did spin up the model for one year. The following text has been inserted to the manuscript L212:

*"The models spun up for one year to ensure the equilibrium status."*

Line 290. If I understand correctly, here should be "parameter", which is different from "parameterization

**Response:** Thanks for pointing it out. We have corrected it.

This paper presents some limited results of evaluating the impact of different formulations in runoff generation schemes when simulating streamflow. My major objection with the paper is that it really is not assessing the uncertainty but rather the variability of the simulated streamflow and how each of the forcings, model parameters or formulations contribute to it. Although that is valuable in itself, the authors claim that the objective is to identify the uncertainties in the context of climate change simulations. However, that is not what was done here. The calibration of the parameters was done using an observation-based forcing dataset and although I can understand the rationale, I believe that any calibration of parameters should have been done in a way that would emulate the intended application (i.e. using GCM output to drive the hydrology model). I believe historical simulation are available from CMIP5 and if so they should be used to evaluate the actual uncertainty of simulated streamflow within the framework that the authors have developed. The end of 21st century simulations should be a final experiment (if included at all) given the objective of the paper. Consequently, I recommend major revisions before publication that will include new simulations that test the different model parameter sets, runoff generation schemes and downscaled GCM output for the period when streamflow measurements are available, so that the actual uncertainty can be quantified. In addition, I believe the study area is rather limited and an opportunity is being missed by not including additional basins with different physiography and climate.

**Response:** In this study, we did use GCM simulations as the forcings of hydrologic models for the historical period. For each simulation scenario (i.e., the combination of hydrologic model, parameter set, GCM and RCP), we simulated runoff and discharge using GCM outputs for both historical and future periods, and then the relative changes (%) between future and historical simulations were quantified. The total uncertainty in these projected changes from all model combinations (3 hydrologic models, 3 parameter sets, 10 GCMs and 2 RCPs; 3*3*10*2=180) was quantified. The uncertainty contributions to the total uncertainty were then quantified using the ANOVA method.

To clarify it, we have inserted the following text to the manuscript:

*"Here, we used GCM outputs as the forcings of hydrologic models for both historical (1986-2005) and future (2081-2100) periods. For each simulation scenario (i.e., the combination of hydrologic model, parameter set, GCM and RCP), the historical and future daily streamflow and monthly runoff were simulated, and the relative changes (%) were quantified."*

To deepen this study, we have also expanded the analysis by including more metrics about the volume and composite of runoff (i.e., monthly surface, subsurface and total runoff), as well as the hydrologic seasonality (wet season length and timing of wet season onset), considering these quantities are of great importance for the study region (Myers et al., 2019;Feng et al., 2019).
We have added the following figures and texts in the manuscript:

[Figure]

*Figure 6: (a) Projected relative changes (%) in monthly surface runoff, subsurface runoff and total runoff in the whole study region during 2081-2100 as compared to historical period (1986-2005); (b) Relative contributions (%) of the uncertainties for the projected changes in the monthly total runoff; Hydro = Hydrologic models; Para = hydrologic model parameters; GCM = General Circulation Models; RCP = Representative concentration pathways (emission scenarios); "other" is the uncertainty from the 3rd and 4th orders of interactions between the 4 major sources (i.e., GCMs, RCPs, Hydrologic models and parameters).*

*"The projected changes in monthly runoff (surface, subsurface and total) during 2081-2100 compared to 1986-2005 range between -100% and 300% (Figure 6a). Surface runoff will probably increase in February and March, and decrease in other months (Figure 6a). This is because in the future, the onset of wet season will be delayed and more severe storm events will*

*occur during the shorter wet season (Feng et al., 2019). The decrease in subsurface runoff in all months is probably because the decrease in the frequency (or total number) of storm events (Feng et al., 2019). The changes of monthly total runoff show similar pattern with the surface runoff, suggesting the more pronounced changes in surface runoff as compared to subsurface runoff. The major uncertainty sources are GCM and RCP, which account for ~45% of total uncertainty (Figure 6b). Hydrologic models contribute to ~10% of total uncertainty (Figure 6b). This suggests that the climate patterns (e.g., storm event frequency and intensity) are more important factors controlling the runoff generation than the hydrologic model algorithms."*

[Figure]

**Figure 9**: **(a)** Projected change (days) in the onset and duration of wet season in SBC; positive (negative) values indicate later (earlier) onset or longer (shorter) duration of the wet season; **(b)** relative contributions (%) of the uncertainties of the projected changes in seasonality. Hydro = Hydrologic models; Para = hydrologic model parameters; GCM = General Circulation Models; RCP = Representative concentration pathways (emission scenarios); "other" is the uncertainty from the 3rd and 4th orders of interactions between the 4 major sources (i.e., GCMs, RCPs, Hydrologic models and parameters).

*"Consistent with the work of Feng et al. (2019), this study suggests a delayed onset and shorter duration of wet season (Figure 9a). The median changes show that the wet season will start later by 3 days, and become shorter by ~6 days. The major uncertainty sources for both onset and duration of wet season are GCM (~20%) and hydrologic models (~15%). Different from discharge and runoff, the seasonality shows more uncertainty from hydrological models (15% vs 12%) and model parameters (~6% vs 2%) (Figure 9b). This is because the seasonality integrates the runoff generation, paths and transport processes for both surface and subsurface runoff, which are important for the timing and quantity of simulated discharge."*

Some additional comments are outlined below:

\* How does the uncertainties in the prescribed ET affect the results? Why weren't they accounted for?

**Response:** We agree that the uncertainty from ET models will affect the results. However, the main focus of this study is to investigate the uncertainty contributions of runoff generation schemes and associated parameters for the climate change impact assessment. Therefore, we used the same ET method and routing algorithm for all three hydrologic models. To clarify it, we have added the following text in the discussion section:

*"Compared to previous studies (e.g., Vetter et al. (2015), Schewe et al. (2014), Hagemann et al. (2013);(Troin et al., 2018), and Asadieh and Krakauer (2017)), this work identifies relatively low uncertainty contributions from hydrologic models. The main reason for this is that the hydrologic model uncertainty in this study was only from runoff generation algorithms and associated parameters. As is, the three hydrologic models share common algorithms for ET and plane/channel routing, and the same model configuration (e.g., soil matrix and model unit definition). These similarities among models likely reduced the differences in simulated runoff and discharge. In addition, the uniform calibration approach and parameter selection criteria were also likely to eliminate user/method bias which is common in studies that consider more than one hydrologic model. In contrast, the hydrologic models used in previous studies have their own model configurations, and ET and routing algorithms. For example, the VIC model (here VIC refers to the original VIC model, and is different from the model used in this study; to clarify, in following text, VIC refers to the original VIC model while VIC-HRR refers to the model used in this study) applies an ET algorithm different from the one used in this study (Raoufi and Beighley, 2017), uses the grid-based model units ignoring the spatial arrangement, and has its own routing scheme which adopts the synthetic unit hydrograph concept. These differences between models likely resulted in the larger uncertainties in the simulation from hydrologic models in previous studies.*

*This study can also provide useful information for selecting hydrologic models for climate change impact analysis. As discussed in section 3.1, the STP-HRR model is more suitable than the other two models for the study region, mainly due to its ability to represent the non-linear hydrological response to precipitation forcings. This implies hydrologic models adopting the saturation excess runoff generation algorithms may be more suitable for areas with a Mediterranean climate. The uncertainties from hydrologic models are larger than those from the hydrologic model parameters for all hydrologic variables (e.g., discharge, runoff and seasonality), suggesting the inter-model variability is larger than the intra-model variability (from model parameters). This implies that model selection is more important than the parameter selection, and that the parameter equifinality (or non-uniqueness) is less of a concern when quantifying climate change impacts on hydrologic fluxes when using an ensemble of GCM forcings. In this study, only the runoff generation algorithm was investigated. Other hydrologic model components, such as ET algorithm and routing method, also have many variants. The choice of these components can also make a difference in the total uncertainties in simulated runoff and streamflow. Therefore, further study integrating different algorithms for these components can be conducted in the future. This complete analysis can be useful to guide*

*stakeholders to select appropriate hydrologic algorithms for climate change impacts analysis and to develop actionable adaptation and mitigation strategies."*

\* Abstract needs some attention, especially after l. 21 in terms of cohesiveness. Right now, it reads as bullet points stitched together. Some proofreading needed for redundant articles and grammatical errors.

**Response:** We have modified the abstract as follows. We have also taken a careful proofreading for the whole manuscript.

*"Assessing the impacts of climate change on hydrologic systems is critical for developing adaptation and mitigation strategies for water resource management, risk control and ecosystem conservation practices. Such assessments are commonly accomplished using outputs from a hydrologic model forced with future precipitation and temperature projections. The algorithms used for the hydrologic model components (e.g., runoff generation) can introduce significant uncertainties in the simulated hydrologic variables. Here, a modeling framework was developed that integrates multiple runoff generation algorithms with a routing model and associated parameter optimizations. This framework is able to identify uncertainties from both hydrologic model components and climate forcings as well as associated parameterization. Three fundamentally different runoff generation approaches: runoff coefficient method (RCM, conceptual), variable infiltration capacity (VIC, physically-based, infiltration excess) and simple-TOPMODEL (STP, physically-based, saturation excess), were coupled with the Hillslope River Routing model to simulate surface/subsurface runoff and streamflow. A case study conducted in Santa Barbara County, California, reveals increased surface runoff in February and March while decrease in other months, a delayed (3 days, median) and shortened (6 days, median) wet season, and increased daily discharge especially for the extremes (e.g., 100-yr flood discharge, Q100). The uncertainties of the projected changes in these hydrologic variables are large (e.g., 400% for monthly runoff and 340% for Q100). For runoff and discharge, general circulation models (GCMs) and emission scenarios are two major uncertainty sources, accounting for about half of the total uncertainty. For the changes in seasonality, GCMs and hydrologic models are two major uncertainty contributors (~35%). In contrast, the contribution of hydrologic model parameterization to the total uncertainty of changes in these hydrologic variables is relatively small (<6%), limiting the impacts of hydrologic model parameter equifinality in climate change impact analysis. This study also provides insights on how to optimize hydrologic model selection for projecting future hydrologic conditions."*

\* l. 53: what is the need for naming the "land-atmosphere interactions" as "runoff generation process" when the latter is clearly one of the processes that manifest from those interactions?

**Response:** We have modified the text to following:

*"Generally, hydrologic models have modules simulating water partitioning at land surface (named as runoff generation process in this study), evapotranspiration, and water transportation along terrestrial hillslopes and channels (named as routing process here)."*

\* l. 175- : Not sure whether this much detail is needed for the description of the runoff generation models, since they are well established.

**Response:** We have moved the text associated with runoff generation models to the supporting information.

* l. 354: does that mean that there is bias in the validation data (i.e. streamflow)?

**Response:** Yes, we think so. The typical uncertainty for streamflow gauge data is 6%-19% in small watershed based on previous studies (e.g., Harmel et al., 2006). The work of Beighley et al. (2003) also identified the bias in the 1995 January event.

We discussed this in L363-368:

*"The uncertainties in gauge measurements can also be a bias source. For example, in typical conditions the uncertainty in streamflow measurements ranges between 6%-19% in small watersheds, but it can be higher during large storm events when accurate stage measurements are more difficult (Harmel et al., 2006). Beighley et al. (2003) also identified the overestimation of gauge records for the 1995 January event at Gauge 11119940."*

* l. 362-363: this highlights another problem that has not been addressed in this study: the downscaling of GCM outputs to drive the hydrology model.

**Response:** We agree that the uncertainty from the downscaling of GCM outputs can impact the results. The focus of this study is the uncertainty contribution from the runoff generation models and associated parameters. Therefore, we didn't include different downscaling methods and quantify their uncertainties. However, we do think it is a great idea, and we may investigate it in our future studies.

References:

Addor, N., Rössler, O., Köplin, N., Huss, M., Weingartner, R., and Seibert, J.: Robust changes and sources of uncertainty in the projected hydrological regimes of Swiss catchments, Water Resources Research, 50, 7541-7562, 10.1002/2014wr015549, 2014.
Asadieh, B., and Krakauer, N. Y.: Global change in streamflow extremes under climate change over the 21st century, Hydrology and Earth System Sciences, 21, 5863, 2017.
Beighley, R. E., Melack, J. M., and Dunne, T.: Impacts of California's climatic regimes and coastal land use change on streamflow characteristics, JAWRA Journal of the American Water Resources Association, 39, 1419-1433, 10.1111/j.1752-1688.2003.tb04428.x, 2003.
Chegwidden, O. S., Nijssen, B., Rupp, D. E., Arnold, J. R., Clark, M. P., Hamman, J. J., Kao, S.-C., Mao, Y., Mizukami, N., Mote, P. W., Pan, M., Pytlak, E., and Xiao, M.: How Do Modeling Decisions Affect the Spread Among Hydrologic Climate Change Projections? Exploring a Large Ensemble of Simulations Across a Diversity of Hydroclimates, Earth's Future, 7, 623-637, 10.1029/2018ef001047, 2019.
Feng, D., Beighley, E., Raoufi, R., Melack, J., Zhao, Y., Iacobellis, S., and Cayan, D.: Propagation of future climate conditions into hydrologic response from coastal southern California watersheds, Climatic Change, 153, 199-218, 10.1007/s10584-019-02371-3, 2019.
Hagemann, S., Chen, C., Clark, D. B., Folwell, S., Gosling, S. N., Haddeland, I., Hanasaki, N., Heinke, J., Ludwig, F., Voss, F., and Wiltshire, A. J.: Climate change impact on available water resources obtained using multiple global climate and hydrology models, Earth Syst. Dynam., 4, 10.5194/esd-4-129-2013, 2013.

Hattermann, F. F., Vetter, T., Breuer, L., Su, B., Daggupati, P., Donnelly, C., Fekete, B., Flörke, F., Gosling, S. N., Hoffmann, P., Liersch, S., Masaki, Y., Motovilov, Y., Müller, C., Samaniego, L., Stacke, T., Wada, Y., Yang, T., and Krysnaova, V.: Sources of uncertainty in hydrological climate impact assessment: a cross-scale study, Environmental Research Letters, 13, 015006, 10.1088/1748-9326/aa9938, 2018.

Myers, M. R., Barnard, P. L., Beighley, E., Cayan, D. R., Dugan, J. E., Feng, D., Hubbard, D. M., Iacobellis, S. F., Melack, J. M., and Page, H. M.: A multidisciplinary coastal vulnerability assessment for local government focused on ecosystems, Santa Barbara area, California, Ocean & Coastal Management, 104921, https://doi.org/10.1016/j.ocecoaman.2019.104921, 2019.

Raoufi, R., and Beighley, E.: Estimating daily global evapotranspiration using penman–monteith equation and remotely sensed land surface temperature, Remote Sensing, 9, 1138, 2017.

Schewe, J., Heinke, J., Gerten, D., Haddeland, I., Arnell, N. W., Clark, D. B., Dankers, R., Eisner, S., Fekete, B. M., Colón-González, F. J., Gosling, S. N., Kim, H., Liu, X., Masaki, Y., Portmann, F. T., Satoh, Y., Stacke, T., Tang, Q., Wada, Y., Wisser, D., Albrecht, T., Frieler, K., Piontek, F., Warszawski, L., and Kabat, P.: Multimodel assessment of water scarcity under climate change, Proceedings of the National Academy of Sciences, 111, 3245-3250, 10.1073/pnas.1222460110, 2014.

Troin, M., Arsenault, R., Martel, J.-L., and Brissette, F.: Uncertainty of Hydrological Model Components in Climate Change Studies over Two Nordic Quebec Catchments, Journal of Hydrometeorology, 19, 27-46, 10.1175/jhm-d-17-0002.1, 2018.

Vetter, T., Huang, S., Aich, V., Yang, T., Wang, X., Krysanova, V., and Hattermann, F.: Multi-model climate impact assessment and intercomparison for three large-scale river basins on three continents, Earth System Dynamics, 6, 17, 2015.

---

## Author Response (AR1)

Dr. Hilary McMillan                                                                                                Jan 7th, 2020
*Editor*
*Hydrology and Earth System Sciences*

Dear Dr. McMillan,

We thank you for your effort in collecting three valuable review reports for our paper entitled "Identifying uncertainties in hydrologic fluxes and seasonality from hydrologic model components for climate change impact assessments" (manuscript ID: HESS-2019-328). We appreciate all reviewers for their solid comments. We have addressed all of them to the extent possible.

Particularly, we have expanded the study by adding analysis about runoff paths (surface and subsurface), magnitude and timing, and hydrologic seasonality (onset and duration of wet season). Dynamics of runoff and discharge are crucial to coastal ecosystems in the study region, as they are major carriers of nutrients/sediment exported to the coast. The nutrients/sediment fluxes are positively correlated with hydrologic variability, and most of them occur at the beginning of the wet season. Therefore, both timing and magnitude of runoff and discharge will impact the nutrients/sediment export. The runoff also impacts soil moisture, which thus affects the occurrence of droughts and wildfires. The findings with the new analysis reveal that the surface runoff and river discharge (especially the extremes) will increase but get delayed during wet season, while decrease during dry season. The uncertainty analysis reveals that GCMs and RCPs are two major uncertainty contributors for changes in runoff and discharge. In contrast, GCMs and hydrologic models are top two uncertainty sources for changes in seasonality. The identified changes in runoff, discharge and seasonality and associated uncertainties have significant implications for practices and studies in many fields, such as water resources, agriculture, ecosystems conservation, and risk control. In addition to the results of the new analysis, we have also added a detailed discussion in the Results and Discussion section.

We have also improved the analysis about the hydrologic model parameters by developing and applying more sophisticated parameter selection procedure. Both parameter dominance and variability have been considered. The results with new parameter sets suggest that the uncertainty contribution of hydrologic model parameters for changes in seasonality is larger than those for runoff and discharge. However, the contribution of hydrologic model parameters is small, as compared to other sources. We have added a detailed discussion about this finding and its implications.

Finally, we have conceptualized this study with previous uncertainty analysis literature. The major contribution of this study is that it is the first study investigating hydrologic model uncertainty solely from runoff generation algorithms for a region with the Mediterranean climate. The framework developed in this study can be potentially used to identify internal uncertainties of hydrologic models, i.e., uncertainties from different model components (e.g., runoff generation algorithms, ET algorithms and routing models). This is particularly important for assessing model performance and enhancing our understanding of relative roles of model components in the uncertainty contribution. The unique climate pattern (i.e., Mediterranean, characterized by dry summers and cool, moist winters) and the highly non-linear relationship

between precipitation and hydrologic fluxes in the study region lead to representative characteristics of hydrology in Mediterranean regions, which significantly impact local society, agriculture and ecosystems. The findings in this study, including the favorability of STP algorithm, the important role of GCM selection and the negligible role of hydrologic model parameters in terms of uncertainty analysis, can be informative for studies associated with hydrologic model assessment/selection and climate change impact analysis for other Mediterranean regions.

We have provided specific responses to each comment in the follow text. We are certain that with these modifications, we have significantly improved the manuscript.

If you have any further questions, please contact me.

Sincerely,

X ________________________

Dongmei Feng

Dongmei Feng, Ph.D.
Department of Civil and Environmental Engineering
University of Massachusetts, Amherst
130 Natural Resources Road, Amherst, MA 01003
Tel: (617) 697-8789
Email: dmei.feng@gmail.com

**RC1**
**General Comments** (overall quality of paper):
Overall, I think the paper was well written as a climate impacts assessment and application of uncertainty methods provided elsewhere. As stated in the Introduction, the goals of the paper were threefold:
1) Compare different hydrologic models 2) Quantify uncertainty associated with different modeling choices 3) Provide suggestions for studies looking at impacts of climate change
The paper accomplished these three goals (aside for one point which I address in the paragraph below). However, the authors don't make it clear how the field is moved forward even if all three goals are achieved. The authors' study appears to be similar to the Vetter et al (2015) study but in a different, and more homogeneous, domain. As-is, the authors conducted a very detailed assessment of climate change impacts on streamflow in Santa Barbara County. Their assignment of uncertainty to different modeling components followed methods similar to those in previous studies like Addor et al (2014), Vetter et al (2015), Hattermann et al (2018), Chegwidden et al (2019). I did not see any truly novel methods proposed, thus making the paper seem more like an, albeit very rigorous, report. As is, the study is appropriate for a climate impacts assessment journal, but the findings are insufficiently new to warrant publication in HESS.
To make the manuscript more relevant to HESS, I suggest a handful of other potential additions to deepen the analysis. Would it be possible to expand the analyses conducted here to other domains and thus do an intercomparison across different regions? For example, the findings in Figures 5 through 7 are relatively uniform across region and metric. Perhaps the authors could probe deeper into those comparisons by conducting more analyses in other regions or with other metrics? By expanding the analysis to other regions and metrics the study could test how sensitive the uncertainty analysis is to the research question of interest.
**Response:** To deepen this study, we have expanded the analysis by adding metrics about runoff paths (surface and subsurface), magnitude and timing, and hydrologic seasonality (onset and duration of wet season). Dynamics of runoff and discharge impact nutrients/sediment transport, soil moisture, occurrence of flash floods, droughts and wildfires in the study region, which are thus crucial to local society, agriculture and coastal ecosystems. The new analysis reveals new interesting findings, including the increase in surface runoff during wet season while decrease in dry season, a decrease in subsurface runoff, a shortened and delayed wet season, and variable roles of hydrologic models and parameters for different variables (runoff, discharge, and seasonality) in terms of uncertainty contribution. We believe these findings will be useful for practices and studies in many fields, such as water resources, agriculture, ecosystems conservation, and risk control. We have also added a detailed discussion about this in the manuscript. The study region is a representative Mediterranean area. The findings in this study, such as the favorability of STP algorithm, the important role of GCM selection, and the negligible role of hydrologic model parameters in uncertainty, can be informative for studies associated with hydrologic model assessment and climate change impact analysis for other Mediterranean regions. We have added the new results (figures and text) and associated discussions in the manuscript.

The corresponding changes are listed as below:

L103-123:

[revised manuscript text omitted]

Another potential avenue of analysis could be a deeper understanding of the parameter space. I am skeptical about the finding that parameterizations explained little of the uncertainty since it appeared (from Figure 4) that the values within the different parameter sets evaluated were actually quite similar. Since it appears that you have those parameter sets available, would it be possible to expand the analysis to include more parameter sets? That could buy more confidence in the current analyses.

**Response:** We have modified the parameter sets analysis by implementing more sophisticated parameter selection criteria and selecting more parameter sets. We first selected 4 parameter sets with highest NSE as we did originally, and then we selected another 6 sets of parameters following the procedures: (1) rank the rest parameter sets based on their performance (assessed by NSE); (2) randomly select 6 sets of parameters from the top 20% samples. These 10 parameter sets take both parameter dominance and variability into account. Then they were used for the uncertainty analysis. The associate figure and results have been updated.

*L220-226:*

*"To quantify the uncertainties from model parameters, we selected 10 parameter sets using the following criteria: (1) select 4 parameter sets with highest NSE based on the calibration results; (2) rank the rest parameter sets based on their performance (i.e., NSE), and randomly select 6 sets from the top 20% candidates. This parameter selection process enabled us to take both parameter dominance and variability into account, while maintaining the high model performance, which is important for the uncertainty analysis. These 10 parameter sets were then used for uncertainty analysis."*

[Figure]

*Figure 5: Parameters (black circles) sampled during calibration process and their corresponding performance (assessed by NSE). The red circles indicate the 4 parameter sets with highest NSE values, and the green circles indicate 6 randomly selected parameter sets from the top 20% samples (ranked by NSE). These ten parameter sets were used for uncertainty analysis. In this figure, the parameter values are normalized by their ranges (shown in Table 1), so the range of the x axis is 0-1. The parameters were sampled throughout their whole ranges, however, for clarity, samples with NSE lower than 0.3 are not shown in this figure.*

References not included in the current manuscript:

Addor, N., Rössler, O., Köplin, N., Huss, M., Weingartner, R., & Seibert, J. (2014). Robust changes and sources of uncertainty in the projected hydrological regimes of Swiss catchments. Water Resources Research, 50(10), 7541–7562. https://doi.org/10.1002/2014WR015549

Chegwidden, O. S., Nijssen, B., Rupp,D. E., Arnold, J. R., Clark, M. P.,Hamman, J. J., et al. (2019). How do modeling decisions affect the spread among hydrologic climate change projections? Exploring a large ensemble of simulations across a diversity of hydroclimates. Earth's Future,7,623–637. https://doi.org/10.1029/2018EF001047

Hattermann, F. F., Vetter, T., Breuer, L., Su, B., Daggupati, P., Donnelly, C., Krysnaova, V. (2018). Sources of uncertainty in hydrological climate impact assessment: A cross-scale study. Environmental Research Letters, 13(1). https://doi.org/10.1088/1748-9326/aa9938

**Response:** They have been inserted to the manuscript in L77-81.

==================================

Specific Comments (individual scientific questions/issues):

L14- In the abstract, you mention that identification and uncertainties are rarely studied. This is not true. It is increasingly common (see, for example, the three references above).

**Response:** With that sentence, we meant the identification of uncertainties from the hydrological model components (not the whole hydrologic model) is rarely studied. We have modified the sentence as follows:

L12-16:

"*Such assessments are commonly accomplished using outputs from a hydrologic model forced with future precipitation and temperature projections. The algorithms used for the hydrologic model components (e.g., runoff generation) can introduce significant uncertainties in the simulated hydrologic variables. Here, a modeling framework was developed that integrates multiple runoff generation algorithms with a routing model and associated parameter optimizations. This framework is able to identify uncertainties…*"

L139-141 – Is the monthly 1 degree aerosol optical depth product sufficient for calculating radiation at the scale you are working at?

**Response:** The 1 degree aerosol optical depth product was downscaled to 0.05 degree for PET estimation. The following text has been inserted to the manuscript:

L149-150:

"*The aerosol optical depth product was downscaled to 0.05° x 0.05° (Raoufi and Beighley, 2017).*"

L154-156 – Should there be some discussion about the fact that, regardless of subbasin size (which, as the authors state, ranges between 0.1 and 135 km^2) the parameters are averaged across each subbasin?

**Response:** The "sub-basin" here refers to the model unit, which was indicated in Lines 162-164 "*The sub-basins are irregular-shape catchments defined by the flow accumulation area threshold. In this study, the threshold is 1 km$^2$, which means the sub-basins (model units) are in size of roughly 1 km$^2$.*" So, the watershed with a drainage area of 135 km$^2$ roughly consists of 135 model units, and the model parameters are averaged over each model unit. For small

watersheds with drainage areas less than 1 km$^2$, the model parameters were averaged over each watershed. To clarify it, we have added the following text to the manuscript:

L166-168:

*"This indicates these parameters were averaged for each model unit, the majority of which had an area of roughly 1 km$^2$, with less than 1% having an area of <1 km$^2$."*

L175-254 – I'm not sure the specificity is necessary for each of the hydrologic models in the main text. I would suggest moving the conceptual plot from the supplemental text to the main text and moving the mathematical explanations to the supplemental text. This would save you space in the main body of the text, improving readability, while letting your story come through easier. With that space you could fill in with more details on the calibration methods.

**Response:** We have moved the text about hydrologic models to the Supporting Information and added a conceptual figure (Figure 2) to illustrate the framework.

[Figure]

*Figure 2: The conceptual framework about the hydrologic models used in this study. Portions of this figure were adapted from the work of Beighley et al. (2009). (a) shows the grid-based climate inputs for hydrologic models; (b) shows water balance models; P is precipitation; ET is evapotranspiration; $E_s$ is soil evaporation; $E_c$ is canopy evaporation; $E_T$ is transpiration; $e_s$ is water available for surface runoff; $e_{ss}$ is water available for subsurface runoff; $\theta_U$ is relative soil moisture in upper soil layer; $\theta_L$ is relative soil moisture in lower soil layer; I is infiltration; K is water flux from the upper layer to the lower layer; and D is diffusive water flux from the lower layer to the upper layer; and (c) shows HRR routing model; the "open-book" assumption: two identical planes (P1 and P2) with the channel (Ch) in the center of each sub-basin; $q_s$ is the surface runoff; $q_{ss}$ is subsurface runoff; Q is discharge in the river channel, and WT is groundwater table. The parameters in red italic are for surface runoff generation, the parameters in blue italic are for subsurface runoff generation. The first columns in the tables indicate the models that the parameters are used for. The definition of these parameters can be found in the supporting information.*

L260-263 – The definitions of Kss and Ks would probably best fit in the description of the routing model since they are from that model.

**Response:** We have moved them to the routing model section L202-205.

L268-269 – How are the three different optimal parameter sets selected? Are they very different parameter sets? As in, are they likely to be in very different parts of the overall calibration space/range of parameter values? Or are they likely to be relatively similar? If they are similar, does that explain the less than 1% uncertainty explained by parameterization referenced in L426 in the results? I see in Figure 4 that some of the parameter values for some models are indeed quite similar (e.g. Kss_all for RCM-HRR). How does this affect your conclusions about the minimal contribution of parameterization toward total uncertainty?

**Response:** We think the reviewer is right that some parameters are quite similar, which explains the small uncertainty contribution. We have modified the parameter selection process by applying the following criteria: (1) select 4 parameter sets with highest NSE based on the calibration results; (2) rank the rest parameter sets based on their performance (i.e., NSE), and randomly select another 6 sets from the top 20% samples. This parameter selection process enabled us to take both parameter dominance and variability into account while maintaining the high model performance. The results with these 10 parameter sets show that for changes in runoff and discharge, the hydrologic model parameters contribute about 2% to the total uncertainty, and for changes in seasonality (onset and duration of wet season), the parameters can contribute ~6% of the total uncertainty. The associated figures (Figures 5-9) and text (section 3.2) have been updated.

L320-324 – Do the authors conduct their performance weighting based upon the GCM simulations? Or do they do it using the historical meteorological forcing data (in this case Livneh et al)? The latter would be appropriate, since the former would not match the actual weather experienced by the region.

**Response:** We did the performance weighting on the simulations from the GCMs+HydroModels+Parameters combinations, because we need to assess their performance in representing historical reality and then assign weights to them. The reviewer is right that GCMs don't match the actual weather. To deal with this issue, we ranked the simulated discharge variable series (e.g., annual mean discharge or annual peak discharge) from each GCMs+HydroModels+Parameters simulation, and then compared them with the observations (also ranked). The details of this process have been provided in the Supporting Information.

"*In this study, the annual mean discharge and annual maximum daily discharge are the considered variables. Since these two variables are not normally distributed, a Box-Cox transformation is performed before applying the Expectation–Maximization method. Considering the GCMs' predictions are not temporally consistent with reality (i.e., the GCMs' prediction does not have correct timing), the observation and simulation are both ranked from high to low, and then $l(\theta)$ is maximized based on the ranked series. The procedure is as follows:*
**Step I:** *Calculate the observed annual mean discharge (or annual maximum daily discharge) at each watershed of interest for the period 1986-2005*
**Step II**: *Calculate the simulated annual mean discharge (or annual maximum daily discharge) for each simulation in the ensemble (3×10×10=300 models) for the same period*

*Step III: Rank the observed and simulated annual mean discharge (or annual maximum daily discharge) in a descending order*
*Step IV: Calculate the Box–Cox coefficient λ for each watershed by using the BoxCox.lambda function in R and transform the quantities by using Eq. S28:*

$$z_t = \frac{y_t^\lambda - 1}{\lambda} \qquad (S28)$$

*Step V: Apply the EM process to the transformed series $z_t$ and estimate the weights and variance of all models*
*Step VI: Calculate the probability of estimated changes in $Q_m$, $Q_p$ and $Q_{100}$ in the future (2081-2100) relative to 1986-2005 using the weights obtained in Step V."*

L360-363 – Is the climate data, even though it was downscaled to the 1/16th degree scale, appropriate for the scale of the subbasins the authors are evaluating (for instance the basin that is only 0.1 km^2)? As the authors suggest in L360-363, they note substantial biases in the precipitation that, in one example case, doesn't even provide enough water to account for streamflow in absence of ET. Did the authors modify precipitation at all to account for this? If not, do the authors think that some other modification of the precipitation forcing would be appropriate? Also, in Figure 2 caption: (a) what does "normalized calibration process" mean?

**Response:** We agree with the reviewer that the 1/16[th] degree precipitation is too coarse for the 0.1 km$^2$ watershed. However, in this study region, such small basins only account for a small fraction of the total area (<1%). We were aware of this issue, so in this study we mainly focused on the results for the major watersheds (area>7 km$^2$) which account for 83% of the total area (Figure 7). In terms of the precipitation scale, we didn't make modifications to it. However, we do think it is a good point that merits further research efforts. In Figure 2 caption, the "normalized calibration process" means the x-axis range (i.e., 0-1) is normalized by the number of iterations during calibration. The termination of the calibration process was determined by an increment threshold (i.e., the improvement step in NSE), so the numbers of calibrations iterations among models were slightly different. For example, for the Mission Creek watershed (USGS gauge NO. 11119750), the numbers of calibration iterations for these three hydrologic models were 1494, 1460 and 1518, respectively. To normalize these numbers to the range of 0-1 for a better presentation, we divided them by 1494, 1460 and 1518, respectively. The following text has been inserted to the Fig.3 caption:

L678-679:

*"the "normalized calibration process" means the x axis range is normalized by the number of iterations during calibration"*

L437-438 – "Changes in Qm, Qp and Q100 are higher under RCP 8.5, but the uncertainties are also higher (Fig. 8), which suggests the uncertainties from RCPs are mainly introduced by RCP 8.5." Could you clarify this statement? I think there may be some conflation in the sources of the uncertainties in this argument. In looking at Figure 8, we see that the distributions are very different between RCP45 and RCP85. However, in your ANOVA formulation, the comparison of the different model choices really just looks at the differences in the means. Thus, attributing the uncertainty to RCP 8.5 can't be made by these figures alone, since you are only comparing two choices. If you are referring to the large standard deviation of the RCP85 distribution, then

that uncertainty contribution would actually be a higher-order interaction of RCP and something else (perhaps GCM?).

**Response:** We think the reviewer is right. We have modified this sentence to the following:

L395-398:

*"Changes in $Q_m$, $Q_p$ and $Q_{100}$ are higher under RCP 8.5, but the uncertainties are also higher (Figure 8), which suggests the higher contribution of RCP 8.5 in the uncertainties of higher-order interactions between RCP and other factors (i.e., GCM, hydrologic model and parameters)."*

L394-396 – I assume the reference to the National Land Data Assimilation System VIC model set-up is the one referenced at the following DOI? (https://doi.org/10.5067/ELBDAPAKNGJ9) If so, it needs a citation and perhaps some explanation as to why this is used as a suitable comparison.

**Response:** Thanks for this information. We have added the reference in the text. We also have inserted the following text to explain the reason for selecting the NLDAS-VIC model outputs as a comparison.

L340-342:

*"The NLDAS-VIC runoff simulations are from the same runoff generation model (i.e., VIC) as used in this work, and have compatible spatial/temporal resolutions to those in this study, which makes it a suitable reference for comparison."*

L446-449/456-457 – How can you justify that model configurations (e.g. irregular catchments or routing schemes) are the reason that hydrologic models played a smaller role in your uncertainty findings?

**Response:** *We have modified the text as follows:*

L425-443:

*"Compared to previous studies (e.g., Vetter et al. (2015), Schewe et al. (2014), Hagemann et al. (2013);Troin et al. (2018), and Asadieh and Krakauer (2017)), this work identifies relatively low uncertainty contributions from hydrologic models. The main reason for this is probably that the hydrologic model uncertainty in this study was only from runoff generation algorithms and associated parameters. As is, the three hydrologic models share common algorithms for ET and plane/channel routing, and the same model configuration (e.g., soil matrix and model unit definition). These similarities among models likely reduced the differences in simulated runoff and discharge. In addition, the uniform calibration approach and parameter selection criteria were also likely to eliminate user/method bias which is common in studies that consider more than one hydrologic model. In contrast, the hydrologic models used in previous studies have their own model component algorithms (e.g., ET and routing algorithms), and model configurations. For example, the VIC model (here VIC refers to the original VIC model, and is different from the model used in this study; to clarify, in following text, VIC refers to the original VIC model while VIC-HRR refers to the model used in this study) applies an ET algorithm different from the one used in this study (Raoufi and Beighley, 2017), uses the grid-based model units ignoring the spatial arrangement, and has its own routing scheme which adopts the synthetic unit hydrograph concept. When comparing models owning their own component*

*algorithms, the differences between models likely resulted in larger uncertainties in the simulation from hydrologic models in previous studies."*

L449-451 – What do the authors mean by "a common calibration approach is also used to eliminate user/method bias"?

**Response:** In this study, we performed the same calibration procedure for all hydrologic models including the same multi-objective optimization algorithm and the same final parameter selection criteria. Compared to the scenarios when different calibration processes and final parameter selection standards are applied, the calibration approach in this study may possibly generate a more consistent result. To make it clear, we have modified the sentence to:

L432-434:

*"...the uniform calibration approach and parameter selection criteria were also likely to eliminate user/method bias which is common in studies that consider more than one hydrologic model."*

L461-462 – Is reducing the uncertainty the goal for an impacts assessment? Would not the goal actually be to reveal the uncertainty present, and thus actually focus on multiple hydrologic models as the authors suggest that their selection accounts for a sizeable portion of the uncertainty space?

**Response:** The uncertainty induced by hydrologic models is due to their limited capability in representing the real hydrological processes. We have multiple options for hydrologic models. Therefore, it is necessary to quantify the uncertainty caused by the hydrologic model choice, which is one of the main objectives of this study. On the other hand, the performances of different hydrologic models are not the same. For example, the results in this study showed that STP performs better than the other methods. This implies we need to treat these models differently. This is another objective of this study: evaluate and compare the performance of hydrologic models with different approaches representing runoff generation process. To make the statement more appropriate, we have modified the text as below:

L444-454:

*"This study can also provide useful information for hydrologic model evaluation and selection. As discussed in section 3.1, the STP-HRR model is more suitable than the other two models for the study region, mainly due to its ability to represent the highly non-linear hydrological response to precipitation forcings. This implies hydrologic models adopting the saturation excess runoff generation algorithms may be more suitable for areas with a Mediterranean climate. The uncertainties from hydrologic models are larger than those from the hydrologic model parameters for all variables (i.e., discharge, runoff and seasonality), suggesting the inter-model variability is larger than the intra-model variability (from model parameters). This implies that model selection is more important than the parameter selection, and that the parameter equifinality (or non-uniqueness) is less of a concern when quantifying climate change impacts on hydrologic fluxes using an ensemble of GCM forcings."*

L471-475 – At the relatively small scale which you are working, how is routing impactful?

**Response:** We think the reviewer is right that the routing for small basins can be not very impactful. We have modified it to the following:

L454-460:

*"In this study, only the runoff generation algorithm was investigated. Other hydrologic model components, such as ET algorithms and routing methods, also have variants. The choice of these components may also make a difference in the total uncertainties in simulated runoff and streamflow. In addition, the methods for GCM downscaling can also contribute to the uncertainty in predicted changes in hydrology. Further study integrating different algorithms for hydrologic model components as well as GCM downscaling methods can be conducted in the future…"*

L483 – How do you define uncertainty of 230%? Is that the range? Or +/- 2 standard deviations? etc):

**Response:** The uncertainty was defined as the range of predicted changes, that is, max change - min change. To clarify this, we have added the following text:

L485-487:

*"(here, uncertainty refers to the range of predicted relative changes among models, that is, from -100% to +220%)"*

L69-70, L81 – Confusing sentences/phrasing
**Response:** We have modified the sentence in L69-70 to *"Model parameter selections based on calibration metrics can result in different optimal parameter values (i.e., parameter equifinality)."*
We have modified the sentence in L81 to *"Most previous studies treated hydrologic models as a whole package. However, hydrologic models consist of multiple components (e.g., runoff generation, ET and routing). These components can be significantly different among models. When considering the hydrologic model as a whole, it is difficult to quantify relative uncertainty contributions from different components."*
=====================================
Technical Corrections (typing errors, etc.)
L43 – "cause" not "causes"
L220 – "matric" not "metric" – there are many other language typos (e.g. L222 "expresses" should be "expressed") sprinkled throughout the text, but I imagine that with another read-through these issues could be resolved.
Overall, there are small language errors throughout the manuscript which the vast majority of the time don't interfere with understanding but are somewhat distracting. A careful reading would help resolve these.
**Response:** *Thanks for pointing them out. A careful proofreading has been made to correct them.*

**RC2**

This manuscript tries to investigate the uncertainties resulting from different hydrological model components when assessing the impacts of climate change on streamflow. To do so, they design a modeling framework that incorporates three runoff yield schemes, one runoff routing scheme, several GCM and RCP. I think the topic is interesting and the manuscript is overall well-prepared. However, I think there are still several issues have to be addressed before considering for publication in HESS.

The authors choose annual mean discharge, annual peak discharge or 100-yr flood discharge to analyze the uncertainties. I doubt if it's meaningful to investigate annual mean values in a 750 km2 catchment. In figure 5 they even investigate changes and uncertainties in much smaller sub-basin. Because I think, according to their methodology, in such a small catchment the annual mean runoff is simply controlled by precipitation and evaporation. On the other hand, when investigate the annual peak values (here it's not clear how they define 'peak' values, from daily or hourly?), the routing may play a more significant role in the timing and magnitude of simulated streamflow. My concern is if the authors can still reach the same conclusions if they use daily streamflow when perform uncertainties analysis because I believe in such small catchment different runoff yield schemes have more effects on daily streamflow instead annual streamflow.

**Response:** The annual mean discharge was defined as the average of daily streamflow in a year. To clarify it, we have inserted the following sentence in the manuscript:

L274-275:

"*Here, the annual mean discharge was defined as the average of daily streamflow in a year.*"

The annual peak discharge was defined as the maximum daily streamflow in a year. It was described in L274: "*annual maximum daily discharge ($Q_p$)*"

I also want to hear opinions from the authors regarding the choose of runoff yield scheme. When perform regional or global simulations using LSM, people usually can only use one runoff yield option, either saturation-excess (e.g. NoahMP, CLM) or infiltration-excess (e.g. VIC). However, when focus on the specific catchment, you can definitely choose a runoff yield scheme that is suitable for the hydrological regime of that catchment. I'm not challenging your work, just want to hear some discussion.

**Response:** This is a good point. One of the main objectives of this study was to "*evaluate and compare the performance of hydrologic models with different approaches representing runoff generation process…*" (L117-119). The results in this study showed that STP performs better than the other two methods. This finding can be informative for future studies associated with hydrologic model selection. We have inserted the following discussion in the manuscript:

L444-462:

*"This study can also provide useful information for hydrologic model evaluation and selection. As discussed in section 3.1, the STP-HRR model is more suitable than the other two models for the study region, mainly due to its ability to represent the highly non-linear hydrological response to precipitation forcings. This implies hydrologic models adopting the saturation excess runoff generation algorithms may be more suitable for areas with a Mediterranean climate. The uncertainties from hydrologic models are larger than those from the hydrologic model parameters for all variables (i.e., discharge, runoff and seasonality), suggesting the inter-model variability is larger than the intra-model variability (from model parameters). This implies that model selection is more important than the parameter selection, and that the parameter equifinality (or non-uniqueness) is less of a concern when quantifying climate change impacts on hydrologic fluxes using an ensemble of GCM forcings. In this study, only the runoff generation algorithm was investigated. Other hydrologic model components, such as ET algorithms and routing methods, also have variants. The choice of these components may also make a difference in the total uncertainties in simulated runoff and streamflow. In addition, the methods for GCM downscaling can also contribute to the uncertainty in predicted changes in hydrology. Further study integrating different algorithms for hydrologic model components as well as GCM downscaling methods can be conducted in the future. Such analysis can be useful to guide stakeholders to select appropriate hydrologic algorithms and to develop actionable adaptation and mitigation strategies to accommodate climate change."*

Line 134-141. The authors use MODIS products to estimate the PET, However, they don't provide any detail regarding how to convert PET into ET for runoff yield simulation. In eq(1)~(7) I don't see any variable related to ET.

**Response:** The ET was extracted from soil at each time step, and then the soil content was updated which was used for water balance calculation in next time step. We have added the following content to the manuscript:

L193-198:

*"The evapotranspiration was estimated using Eq. S15…. After the water fluxes (runoff, ET and water movement between soil layers) were determined, the soil moisture was updated which would be used for the water balance calculation in the next time step."*

Supporting information:

$$ET = min(PET, W - W_{min}) \tag{S15}$$

*where PET is the potential evapotranspiration estimated using the method proposed by Raoufi and Beighley (2017); W is water content in the upper soil layer; $W_{min}$ is the minimum water content in the soil, defined as $0.15 \times W_s$; $W_s$ is soil water content as saturation.*

Line 255. The authors calibrate several parameters related to runoff. But they don't document how they fix the value of soil depth, from dataset or by calibration. In Line 233 they state that the soil depth is based on a previous study but I don't see any description in (Feng et al., 2019). In

their modeling framework, they use quite simple water balance scheme to account for the soil water movement, in this case the soil depth is an important variable determining the soil water holding capacity.

**Response:** The soil depth data was originally from the Soil Survey Geographic (SSURGO) Data Base for Santa Barbara County. This reference has been added to the manuscript L190-191.

Line 256. sim-topmodel uses groundwater depth to calculate runoff yield. Do you spin up the model to reach the equilibrium state?

**Response:** Yes. We did spin up the model for one year. The following text has been inserted to the manuscript.

L209-210:

*"The models spun up for one year to ensure the equilibrium status."*

Line 290. If I understand correctly, here should be "parameter", which is different from "parameterization

**Response:** Thanks for pointing it out. We have corrected it.

**SC1**

This paper presents some limited results of evaluating the impact of different formulations in runoff generation schemes when simulating streamflow. My major objection with the paper is that it really is not assessing the uncertainty but rather the variability of the simulated streamflow and how each of the forcings, model parameters or formulations contribute to it. Although that is valuable in itself, the authors claim that the objective is to identify the uncertainties in the context of climate change simulations. However, that is not what was done here. The calibration of the parameters was done using an observation-based forcing dataset and although I can understand the rationale, I believe that any calibration of parameters should have been done in a way that would emulate the intended application (i.e. using GCM output to drive the hydrology model). I believe historical simulation are available from CMIP5 and if so they should be used to evaluate the actual uncertainty of simulated streamflow within the framework that the authors have developed. The end of 21st century simulations should be a final experiment (if included at all) given the objective of the paper. Consequently, I recommend major revisions before publication that will include new simulations that test the different model parameter sets, runoff generation schemes and downscaled GCM output for the period when streamflow measurements are available, so that the actual uncertainty can be quantified. In addition, I believe the study area is rather limited and an opportunity is being missed by not including additional basins with different physiography and climate.

**Response:** In this study, we did use GCM simulations as the forcings of hydrologic models for the historical period. For each simulation scenario (i.e., the combination of hydrologic model, parameter set, GCM and RCP), we simulated runoff and discharge using GCM outputs for both historical and future periods, and then the relative changes (%) between future and historical simulations were quantified. The total uncertainty in these projected changes from all model combinations (3 hydrologic models, 10 parameter sets, 10 GCMs and 2 RCPs; 3*10*10*2=600) was quantified. The uncertainty contributions to the total uncertainty were then quantified using the ANOVA method.

To clarify it, we have inserted the following text to the manuscript:

L230-235:

"*Here, we used GCM outputs as the forcings of hydrologic models for both historical (1986-2005) and future (2081-2100) periods. For each simulation scenario (i.e., the combination of hydrologic model, parameter set, GCM and RCP), the historical and future daily streamflow and monthly runoff were simulated, and the relative changes (%) were quantified. Note, there is no RCPs for historical period, and we used the same historical simulation for RCP 4.5 and 8.5.*"

To deepen this study, we have also expanded the analysis by including more metrics about the volume and composite of runoff (i.e., monthly surface, subsurface and total runoff), as well as the hydrologic seasonality (wet season length and timing of wet season onset), considering these quantities are of great importance for the study region (Myers et al., 2019;Feng et al., 2019).

We have added the following figures and texts in the manuscript:

[revised manuscript text omitted]

Some additional comments are outlined below:

\* How does the uncertainties in the prescribed ET affect the results? Why weren't they accounted for?

**Response:** We agree that the uncertainty from ET models will affect the results. However, the focus of this study is to investigate the uncertainty contributions of runoff generation schemes and associated parameters for the climate change impact assessment. Therefore, we used the same ET method and routing algorithm for all three hydrologic models. To clarify it, we have added the following text in the discussion section:

L425-460:

*"Compared to previous studies (e.g., Vetter et al. (2015), Schewe et al. (2014), Hagemann et al. (2013);Troin et al. (2018), and Asadieh and Krakauer (2017)), this work identifies relatively low uncertainty contributions from hydrologic models. The main reason for this is probably that the hydrologic model uncertainty in this study was only from runoff generation algorithms and associated parameters. As is, the three hydrologic models share common algorithms for ET and plane/channel routing, and the same model configuration (e.g., soil matrix and model unit definition). These similarities among models likely reduced the differences in simulated runoff and discharge. In addition, the uniform calibration approach and parameter selection criteria were also likely to eliminate user/method bias which is common in studies that consider more than one hydrologic model. In contrast, the hydrologic models used in previous studies have their own model component algorithms (e.g., ET and routing algorithms), and model configurations. For example, the VIC model (here VIC refers to the original VIC model, and is different from the model used in this study; to clarify, in following text, VIC refers to the original VIC model while VIC-HRR refers to the model used in this study) applies an ET algorithm different from the one used in this study (Raoufi and Beighley, 2017), uses the grid-based model units ignoring the spatial arrangement, and has its own routing scheme which adopts the synthetic unit hydrograph concept. When comparing models owning their own component algorithms, the differences between models likely resulted in larger uncertainties in the simulation from hydrologic models in previous studies.*

*This study can also provide useful information for hydrologic model evaluation and selection. As discussed in section 3.1, the STP-HRR model is more suitable than the other two models for the study region, mainly due to its ability to represent the highly non-linear hydrological response to precipitation forcings. This implies hydrologic models adopting the saturation excess runoff generation algorithms may be more suitable for areas with a Mediterranean climate. The uncertainties from hydrologic models are larger than those from the hydrologic model parameters for all variables (i.e., discharge, runoff and seasonality), suggesting the inter-model variability is larger than the intra-model variability (from model parameters). This implies that model selection is more important than the parameter selection, and that the parameter equifinality (or non-uniqueness) is less of a concern when quantifying climate change impacts on hydrologic fluxes using an ensemble of GCM forcings. In this study, only the runoff generation algorithm was investigated. Other hydrologic model components, such as ET algorithms and routing methods, also have variants. The choice of these components may also make a difference in the total uncertainties in simulated runoff and streamflow. Further study integrating different algorithms for these components can be conducted in the future. Such analysis can be useful to guide stakeholders to select appropriate hydrologic algorithms and to develop actionable adaptation and mitigation strategies to accommodate climate change."*

* Abstract needs some attention, especially after l. 21 in terms of cohesiveness. Right now, it reads as bullet points stitched together. Some proofreading needed for redundant articles and grammatical errors.

**Response:** We have modified the abstract as follows. We have also taken a careful proofreading for the whole manuscript.

L21-34:

*"…A case study conducted in Santa Barbara County, California, reveals increased surface runoff in February and March while decreased runoff in other months, a delayed (3 days, median) and shortened (6 days, median) wet season, and increased daily discharge especially for the extremes (e.g., 100-yr flood discharge, Q100). The Bayesian Model Averaging analysis indicates the probability of such increase can be up to 85%. For projected changes in runoff and discharge, general circulation models (GCMs) and emission scenarios are two major uncertainty sources, accounting for about half of the total uncertainty. For the changes in seasonality, GCMs and hydrologic models are two major uncertainty contributors (~35%). In contrast, the contribution of hydrologic model parameters to the total uncertainty of changes in these hydrologic variables is relatively small (<6%), limiting the impacts of hydrologic model parameter equifinality in climate change impact analysis. This study provides useful information for practices associated with water resources, risk control and ecosystems conservation and for studies related to hydrologic model evaluation and climate change impact analysis for the study region as well as other Mediterranean regions."*

* l. 53: what is the need for naming the "land-atmosphere interactions" as "runoff generation process" when the latter is clearly one of the processes that manifest from those interactions?

**Response:** We have modified the text to following:

L54-57:

*"Generally, hydrologic models have modules simulating water partitioning at land surface (named as runoff generation process in this study), evapotranspiration, and water transportation along terrestrial hillslopes and channels (named as routing process here)."*

\* l. 175- : Not sure whether this much detail is needed for the description of the runoff generation models, since they are well established.

**Response:** We have moved the text associated with runoff generation models to the supporting information, and added a figure (Figure 2) to illustrate the modeling framework.

\* l. 354: does that mean that there is bias in the validation data (i.e. streamflow)?

**Response:** Yes, we think so. The typical uncertainty for streamflow gauge data is 6%-19% in small watershed based on previous studies (e.g., Harmel et al., 2006). The work of Beighley et al. (2003) also identified the bias in the 1995 January event.

We discussed this in L306-311:

*"The uncertainties in gauge measurements can also be a bias source. For example, in typical conditions the uncertainty in streamflow measurements ranges between 6%-19% in small watersheds, but it can be higher during large storm events when accurate stage measurements are more difficult (Harmel et al., 2006). Beighley et al. (2003) also identified the overestimation of gauge records for the 1995 January event at Gauge 11119940."*

\* l. 362-363: this highlights another problem that has not been addressed in this study: the downscaling of GCM outputs to drive the hydrology model.

**Response:** We agree that the uncertainty from the downscaling of GCM outputs can impact the results. The focus of this study is the uncertainty contribution from the runoff generation models and associated parameters. Therefore, we didn't include different downscaling methods and quantify their uncertainties. However, we do think it is a great idea, and we may investigate it in our future studies. To clarify it, we have added the following text in the manuscript:

L457-460:

*"...In addition, the methods for GCM downscaling can also contribute to the uncertainty in predicted changes in hydrology. Further study integrating different algorithms for hydrologic model components as well as GCM downscaling methods can be conducted in the future."*

---

## Editor Decision (ED1)

The quantification of internal variability and model uncertainty sources in Multi-scenario Multi-model Ensembles of climate experiments (MMEs) is a key issue. It is expected to both help decision makers to identify robust adaptation measures and scientists to identify where their efforts are needed to narrow uncertainty and give more robust projections. A number of publications have been devoted to this uncertainty issue in the recent years. In hydrological impact studies, the contribution of model uncertainty due to hydrological models has been considered in a few works. Hydrological modelling uncertainty has been found to be a major uncertainty sources for a number of hydrological variables, such as low flows (e. Vidal et al. 2016; Guintoli et al. 2018, Alder et al. 2019). The results presented in previous works however focused on a few specific regions with different specific hydrological behaviours and hydrological processes. Other such studies for other contexts are thus definitively required to gain a better knowledge of the configurations where hydrological modelling is a key contributor to uncertainty in projections.

The present work participates to this effort considering 28 small catchments in California. Its objective is thus really relevant. However it lacks from major limitations and need major improvements before it may be accepted for publication. In short they are :

-       It considers as uncertainty source related to hydrological modelling 2 components of hydrological models: uncertainty due to the "runoff production scheme" and to the "river discharge routing function". The second one is likely not really relevant. A major uncertainty source in hydrological model is on the other hand not accounted for: the one related to the representation of evapotranspiration losses within the catchment

-       The uncertainty analysis framework is not relevant as it disregards the internal variability in climate projections. In addition, the Vetter et al. (2015) subsampling scheme for the ANOVA is not relevant.

-       Some choices retained for hydrological modelling are not really convincing. Among others :

o   The Hydrological Model based on the RunoffCoefficient concept cannot be consider as a relevant hydrological model. Such a representation never corresponds to a modern and state-to-the art representation of hydrological processes.

o   The kinematic wave used to model the subsurface flow in the VIC model is spurious.

-       The Bayesian Model Averaging method to produce pdf"s of change is based on weak assumptions.

The overall description of the work should be otherwise largely improved, especially what is related to the description of hydrological models which have been already described in a number of previous works.

My detailed comments are given below.

Hydrological models and contribution to uncertainty

Low performance models

The performance of the three models is not acceptable. (cf. results obtained for 1/3 of simulated years (figure 2/d)). The representation of the annual water balance is not relevant and has to be improved in all models. How are represented the evapotranspiration losses ? What is the uncertainty related to precipitation inputs? How many precip. stations are available for each catchment ? This may be a serious limitation of the different models considered here.

A low performance model (for the reproduction of observed time series when meteorological forcing data are observed ones) can obviously not be integrated in the analysis. The performance of the RCM model (based on the runoff coefficient) is too low. This is also the case for the VIC Model parametrized here. In such a configuration, as 2 models are known / found to be very poor in the representation of the target system, the Bayesian Model Averaging cannot be considered as a solution to weights the outputs of the different models.

Non relevant RCM model. The Runoff Coefficient Model used for the simulation of the "runoff production" process is based on the simplistic and non-relevant hypothesis that the runoff can be estimated for any time based on a constant runoff coefficient This representation is obviously non-appropriate. (the runoff coefficient actually takes two values depending on the level of relative moisture of a said upper layer but this is obvioulsy not a satisfying / state to the art approach). This model is much too simplistic and cannot be integrated in a state-to-the art uncertainty analysis. If climatologist may consider such a model as acceptable (probably the reason why the authors could publish this model in the "climatic change" journal), no serious hydrologist would consider such a model relevant for the long-term simulation (continuous simulation over multiple years) of the runoff generation processes and then discharge generation processes. This RCM should therefore be either removed from the analysis or adapted (the "runoff coefficient" should at least be a continuous increasing function of the 'relative moisture" variable).

Description of models The descriptions of the models have to be synthetized, have to refer to the original papers which first described the models. In the present analysis context : it is then important to highlight what key representations are different from one model to the other.

Transfer functions. Transfer of runoff and subsurface flow: the kinematic wave is used to simulate both transfer component. It is likely not relevant for the subsurface flow. Is there any reference to give that uses such a model for subsurface flow ? Most models are either physically based, based on Darcy richards equations or are based on conceptual (linear on non linear) transfer reservoirs…. It is also spurious that it could be the official representation in VIC.

Routing model : In the introduction, the routing model is said to be a important contributor of hydrological uncertainty. This is likely not the case as the routing process (and not the runoff transfer processes) in small catchments has little chance to influence significantly flood regimes. It only produces a small distortion of runoff discharge series along the river network. The uncertainty due to the routing model is a priori not an issue. There is thus no reason to introduce this issue in the introduction (by the way, the authors conclude in the introduction they will not consider this issue in the paper, another reason to not introduce this issue there. In all cases, the justification the authors give for this omission (they say "there are less variants in the routing scheme") is not a good reason to disregard this issue. The good reason is mentioned above > in the hydrological behaviour of catchments, the main uncertainty sources are not due to routing but to the production/transfer processes and their representations.

Runoff representation. As mentioned in the introduction, hydrological models usually focus on a given runoff generation process (either Hortonian or Dunian). The runoff generation process is however not the main source of uncertainty. Runoff production is obviously also dependent on the initial saturation conditions of the catchment. Then, the representation of the water balance of the soil and the way its temporal dynamic is represented is important. Appropriate experiments have thus to be found. There are here two 2 majors issues : the type of runoff production (rainfall to rapid

runoff) and the losses by evapotranspiration / evaporation that determine the state of soil storage/saturation… The second one is at least as important as the type of runoff production process (excess saturation / excess infiltration)…. This has to be discussed / integrated in appropriate experiment. …

The way Low Flow are simulated and underground storage is represented could be also an issue. There is likely a large impact on underground storage on the sensitivity of low flows to climatic changes (especially to different changes in different seasons and joint precipitation / temperature changes).

Uncertainty analysis.

·     A deep review of all existing works focusing on the characterization of uncertainty sources in hydroclimate projections is definitively missing. The main conclusions of such works have to be clearly identified. The necessity/interest for additional work on this issue also. The contribution of the present work also. The usually large contribution of internal variability also.

·     Methodology framework used for the Uncertainty Analysis. Authors argue they develop a framework to analyse uncertainty sources related to different sources. They just apply a methodology already presented elsewhere. The way this study could be used for the definition of a proper "methodological analysis framework" would have to be specified.

·     The classical setup of Multimodel Multiscenario of climate experiments makes the number of scenarios, the number of climate models, the number of impact models very different. Vetter et al. 2015. proposed a resampling approach to apply the ANOVA on subsamples of the MME with same numbers for each uncertainty source. This approach is not really relevant as it does not use all available data for a unique / joint estimation of all uncertainty components. Classical ANOVA approaches account for these unbalanced configurations can be used. They typically propose unbiased estimators of all uncertainty components. Refer to any ANOVA handbook.

·     Uncertainty sources accounted for in the analysis disregard internal variability. As in many hydrological impact studies unfortunately, internal variability in hydroclimate projections resulting from the internal variability in climate (leading to low-frequency fluctuations in climate projections) is disregarded. This makes the uncertainty analysis not sound (the GCM uncertainty estimated in the present work is a mix of GCM model uncertainty AND GCM internal variability). This internal variability component is one major uncertainty sources > the authors can not disregard this. The review given in the introduction has to discuss this issue and the framework used to characterize uncertainty sources has to account for this uncertainty source. The authors have especially to read and account for the papers of Hawkins and Sutton, 2011, Hingray and Said, 2014, Hingray et al. 2019 and Evin et al. 2019 for time series ANOVA approaches that are relevant for such configurations (The paper of Hingray et al. 2019 show that the classical Single Time approach of Yip et al. 2010 widley used in the recent years likely produces biased and wrong estimates of most uncertainty components).

PDFs of climate projections

The "Bayesian Model Averaging" work in this work is not appropriate for at least two reasons:

·      It is not relevant to apply it on a set of chains where some are very poorly performing (here hydro models, their performance to reproduce time series of observations from meteorological observation is very poor)

·      Because of internal variability, the evaluation of chains is likely not relevant. I understand that : All combinations of hydrologic models, parameter sets and GCMs (3x3x10=90) seem to have been evaluated. What are the criterion / variables / data periods used for the evaluation ? Do you consider that the evaluation data to be used for each simulation chain are those obtained with this chain for a given control period ? Do you estimate for instance the ability of a given chain to reproduce over this control period the statistical distribution of the observed variable for this control (i.e. the same) control period ? If, yes, this evaluation is not really relevant. The observations are actually one realisation of the recent (control) climate (because of the internal variability) and the simulations are also one realisation of the recent (Control) Climate. So there is no reason why simulations from a given GCM/HydroModel should correspond to observation for a given Ctrl period.

Other comments :

Hydrological models

VIC : The description is not clear. For instance the following 2 sentences have to be clarified: "Ds is the fraction of Dm at which the non-linear base flow begins. Ws is the faction saturation of at which the non linear base flow occurs " … What is the difference between Ds and Ws ? At least, both seem to be very correlated. What is the relation ?

Topmodel :

·      The detailed description of the model is not required. One has to refer to the reference and original paper describing the principle of TOPMODEL (Beven et al .2000)

·      In the current paper, it is not clear on which units TOPMODEL is applied. In the original version, the model is applied in a lumped way, with one single model for the whole basin. Is the model here applied in the same way or is it applied on each hydrological unit ? in the latter case, on which DEM data (resolution) are estimated the topographical index ?

Ln 231 : the water movement between soil layers is similar to that in the modified VIC. Is it for the Topmodel? In all cases, a schematic representation fo the different models (with identification of reservoirs could be added to have a clear idea of the structure of the different models). How many components of discharge are simulated ? runoff + subsurface flow + base flow ??? How is simulated the transfer of each component to the outlet of the catchment ?

Ln 231 – 241 : what is the model described here ?

Ln 258. Give a table with a list of parameters to be calibrated.

What are Ksall and KssAll. Please clarify their role. It is mentioned they allow to account for spatial variability of parameters within the catchment. How ?

How are estimated the parameters of the models for the 28 bassins shown in Figure 5

Equations 29/30 and 31/32 are the same than equations 27/28. To be simplified.

Figure 2a : what is described here ?

References to be considered / integrated:

Importance of hydrological processes in hydrological changes.

Folton, N., Martin, E., Arnaud, P., L'Hermite, P., and Tolsa, M.: A 50-year analysis of hydrological trends and processes in a Mediterranean catchment, Hydrol. Earth Syst. Sci., 23, 2699–2714, https://doi.org/10.5194/hess-23-2699-2019, 2019.

Robust ANOVA frameworks to partition uncertainty sources in MMEs

Hawkins et Sutton (2009). The potential to narrow uncertainty in regional climate predictions. Bulletin of the American Meteorological Society, 90(8), 1095–1108.

Hawkins et Sutton (2011). The potential to narrow uncertainty in projections of regional precipitation change. Climate Dynamics, 37(1-2), 407–418.

Hingray et Saïd 2014. Partitioning internal variability and model uncertainty components in a multimodel multireplicate ensemble of climate projections. J.Climate. 27(17); pp. 6779-6798.

Hingray, et al. 2019. Uncertainty components estimates in transient climate projections. Precision of estimators in the single time and time series approaches. Clim.Dyn. 53:2501-2516

Evin, et al. 2019. Partitioning uncertainty components of an incomplete ensemble of climate projections using data augmentation. J.Climate.

Applications with hydrological model uncertainty

Vidal, et al. 2016. Hierarchy of climate and hydrological uncertainties in transient low flow projections. Hydrol. Earth Syst. Sci.

Giuntoli et al. 2018. Uncertainties in projected runoff over the conterminous United States. Climatic Change.

Alder et Hostetler. 2019. The Dependence of Hydroclimate Projections in Snow-Dominated Regions of the Western United States on the Choice of Statistically Downscaled Climate Data. WRR. 2019.

---

## Author Response (AR2)

Dr.  Hilary McMillan                                                    Mar 17[th], 2020
*Editor*
*Hydrology and Earth System Sciences*

Dear Dr. McMillan,

We thank you for your effort in collecting the feedback report from the reviewer for our revised manuscript (manuscript ID: HESS-2019-328). We have addressed the minor issue identified by the reviewer. We also briefly respond to the comments in the late review report. The detailed responses are provided in the follow text.

If you have any further questions, please contact me.

Sincerely,

X
Dongmei Feng

Dongmei Feng, Ph.D.
Department of Civil and Environmental Engineering
University of Massachusetts, Amherst
130 Natural Resources Road, Amherst, MA 01003
Tel: (617) 697-8789
Email: dmei.feng@gmail.com

**RC1**

**Minor Comment:**

The authors have revised their manuscript and addressed most of my concerns. But they still didn't answer why they only use annual runoff serials for analysis because at such a small scale catchment, using annual runoff can miss many details that may result in different conclusions. From my experience, they should at least use daily runoff. So I want to hear from the authors why they only use annual runoff.

**Response:** In the study region, there is no rain for most time of a year, and it is not uncommon in such a Mediterranean climate region that the annual runoff is mainly generated from one major storm event. Therefore, the annual mean/max series are representative of the characteristics of the discharge dynamics. In fact, we conducted the uncertainty analysis for the daily series, and a similar result was obtained. We have inserted the following text in the manuscript (L277-280):

*"In this study region, there is typically no rain for most time of a year, and it is not uncommon in such a Mediterranean climate region that the annual runoff is mainly generated from one major storm event. Therefore, the annual mean/max series are representative of the characteristics of the discharge dynamics."*

**Response to the late review**

**Response:** We appreciate the comments in the late review report. We have made some changes accordingly. The changes are in L77-81, L95-96, L162-163, and L178-179, and highlighted in red.